# Leaf Nitrogen Concentration and Plant Height Prediction for Maize Using UAV-Based Multispectral Imagery and Machine Learning Techniques

**Lucas Prado Osco** [1,*] , **José Marcato Junior** [2] , **Ana Paula Marques Ramos** [3] ,
**Danielle Elis Garcia Furuya** [3], **Dthenifer Cordeiro Santana** [4], **Larissa Pereira Ribeiro Teodoro** [5],
**Wesley Nunes Gonçalves** [2,6] , **Fábio Henrique Rojo Baio** [5] , **Hemerson Pistori** [6,7] ,
**Carlos Antonio da Silva Junior** [8] **and Paulo Eduardo Teodoro** [4,5]

[1] Faculty of Engineering and Architecture and Urbanism, University of Western São Paulo (UNOESTE), Rodovia Raposo Tavares, km 572 - Limoeiro, Pres. Prudente, SP 19067-175, Brazil

[2] Faculty of Engineering, Architecture and Urbanism and Geography, Federal University of Mato Grosso do Sul (UFMS), Cidade Universitária, Av. Costa e Silva, Pioneiros, MS 79070-900, Brazil; jose.marcato@ufms.br (J.M.J.); wesley.goncalves@ufms.br (W.N.G.)

[3] Post-Graduate Program of Environment and Regional Development, University of Western São Paulo (UNOESTE), Rodovia Raposo Tavares, km 572 - Limoeiro, Pres. Prudente, SP 19067-175, Brazil; anaramos@unoeste.br (A.P.M.R.); daniellegarciafuruya@gmail.com (D.E.G.F.)

[4] Programa de Pós-Graduação em Agronomia - Área de Concentração: Produção Vegetal, da UEMS, Unidade Universitária de Aquidauana, Aquidauana, MS 79200-000, Brazil; dthennyfer.santana@hotmail.com (D.C.S.); paulo.teodoro@ufms.br (P.E.T.)

[5] Department of Agronomy, Federal University of Mato Grosso do Sul (UFMS), Rodovia MS 306, km. 305 Caixa Postal 112, Chapadão do Sul, MS 79560000, Brazil; larissa_ribeiro@ufms.br (L.P.R.T.); fabiobaio@ufms.br (F.H.R.B.)

[6] Faculty of Computing, Federal University of Mato Grosso do Sul (UFMS), Cidade Universitária, Av. Costa e Silva, Pioneiros, MS 79070-900, Brazil; pistori@ucdb.br

[7] Inovisão, Universidade Católica Dom Bosco, Av. Tamandaré, 6000, Campo Grande, MS 79117-900, Brazil

[8] Department of Geography, State University of Mato Grosso (UNEMAT), Av. dos Ingas, 3001 - Jardim Imperial, Sinop, MT 78555-000, Brazil; carlosjr@unemat.br

* Correspondence: lucasosco@unoeste.br

**Abstract:** Under ideal conditions of nitrogen (N), maize (*Zea mays* L.) can grow to its full potential, reaching maximum plant height (PH). As a rapid and nondestructive approach, the analysis of unmanned aerial vehicles (UAV)-based imagery may be of assistance to estimate N and height. The main objective of this study is to present an approach to predict leaf nitrogen concentration (LNC, g kg$^{-1}$) and PH (m) with machine learning techniques and UAV-based multispectral imagery in maize plants. An experiment with 11 maize cultivars under two rates of N fertilization was carried during the 2017/2018 and 2018/2019 crop seasons. The spectral vegetation indices (VI) normalized difference vegetation index (NDVI), normalized difference red-edge index (NDRE), green normalized difference vegetation (GNDVI), and the soil adjusted vegetation index (SAVI) were extracted from the images and, in a computational system, used alongside the spectral bands as input parameters for different machine learning models. A randomized 10-fold cross-validation strategy, with a total of 100 replicates, was used to evaluate the performance of 9 supervised machine learning (ML) models using the Pearson's correlation coefficient (r), mean absolute error (MAE), coefficient of regression (R$^2$), and root mean square error (RMSE) metrics. The results indicated that the random forest (RF) algorithm performed better, with r and RMSE, respectively, of 0.91 and 1.9 g.kg$^{-1}$ for LNC, and 0.86 and 0.17 m for PH. It was also demonstrated that VIs contributed more to the algorithm's performances than individual spectral bands. This study concludes that the RF model is appropriate to predict both

agronomic variables in maize and may help farmers to monitor their plants based upon their LNC and PH diagnosis and use this knowledge to improve their production rates in the subsequent seasons.

**Keywords:** UAV; random forest; nitrogen; maize

## 1. Introduction

Remote sensing techniques aligned with precision agriculture practices are being investigated in researches with different farmlands [1]. In recent years, the increase of market-availability of unmanned aerial vehicles (UAV) encouraged multiple applications in this field. Agriculture remote sensing is a promising field as it supports a multidisciplinary view of different problems related to crop mapping [2] and has been implemented in multiple subjects, such as environment control [3], temporal analysis [4], phenology [5], yield-prediction [6–9], and nutritional analysis [10–12]. These studies revealed the importance of evaluating techniques and sensing data to deal with such tasks.

A relevant topic for farmers and technicians is the correct monitoring of their farmlands, as nutrient absorption rates are connected with plant-growth and yield estimates. An important nutrient related to plant-growth is Nitrogen (N). N benefits leaf development and photosynthetic activity in plants, influencing their productivity [13]. Plants that have nutritional deficiencies related to N show visual symptoms in their leaves, known as chlorosis [14,15]. This nutrient is commonly applied in agricultural areas and it is one of the most contributive nutrients to global production. However, the incorrect diagnosis may be a problem from both economic and environmental point-of-views [16,17].

To circumvent the aforementioned problem, agronomic technicians rely on traditional methods of chemical leaf tissue analysis to determine the amount of N absorbed by the plant [18]. However, this practice is viewed as a destructive, time-consuming, and highly-priced approach. Thus, it is difficult to adopt the traditional analysis as a recurrent procedure to monitor multiple areas and stages [19]. As a rapid, nondestructive, and highly-replicable method, UAV-based image analysis may be of assistance to perform plant nutrient content and growth-status estimate [20–24].

As an alternative method, multispectral data analysis collected with sensor systems represents a promising approach to increase the precision in area monitoring [20]. Predicting nutrient content and plant height with remote systems and automated intelligent methods is gaining attention in agriculture practices. With multispectral sensors, at canopy or leaf levels, different studies predicted leaf nitrogen concentration (LNC) in maize (*Zea mays* L.) [16], winter-wheat (*Triticum aestivum*) [21], cotton (*Gossypium hirsutum*) [22], rice (*Oryza sativa*) [23], citrus (*Citrus sinensis*) [18,24], among others. Although hyperspectral sensors stand out in their ability to characterize the spectral response with high accuracies [25,26], multispectral sensors are used more frequently in agriculture remote sensing since they are economically viable and accessible to most of the front-end users.

Predicting agronomic variables with multispectral data is a common practice in remote sensing applications. However, performing this task with machine learning techniques is still a recent and relevant topic in agriculture remote sensing since it provides a robust and direct approach to evaluate different agronomic variables. Machine learning is considered a subgroup inside of the artificial intelligence area in which algorithms can learn from data and then discover patterns in the dataset, deciding on new and similar information. The algorithms have the potential to model several types of datasets using linear and parametric and nonlinear and nonparametric approaches [12,27,28], including multispectral images [29]. Different machine learning algorithms like random forests (RF), decision trees (DT), artificial neural network (ANN), support vector machines (SVM), among many others, have been adopted to attend various applications in agriculture remote sensing [5,30–32].

Machine learning has helped to increase not only the prediction's accuracy of some agronomic variables but also assisted in solving complex problems related to data heterogeneity. A revision study on yield and N content prediction [33] concluded that advances in remote sensing technologies

and machine learning techniques will result in more cost-effective and comprehensive solutions for a better crop state assessment. The combination of machine learning techniques and vegetation indices (VI) is also an important subject in agricultural applications and has been adopted in different studies, some of which are related to maize characteristics predictions [34,35].

Under ideal conditions of N, maize plants can grow to their full potential reaching maximum height [36,37]. Considering that, implementing different approaches to estimate height and N with UAV-based remote systems is essential to optimize the monitoring of areas with multiple varieties. Currently, one of the main objectives of maize breeding programs is to identify genotypes with high efficiency in N usage [38,39]. Obtaining rapid predictions with an alternative approach like machine learning and UAV-based image may enable programs, technicians, and farmers to evaluate multiple genotypes each year, allowing them to optimize the selection of the most promising plants concerning N use efficiency. In this matter, the main idea behind this proposal is to present a feasible alternative to monitor N and plant height (PH) with machine learning techniques in UAV based imagery.

By implementing the aforementioned approach, farmers can monitor their LNC in maize plants and select the areas or maize varieties (based upon their location or plots) that are most promising based upon their diagnosis and use this knowledge to improve their production rates in subsequent seasons. As machine learning has been proved [23–30] to be a robust approach to evaluate heterogeneous data, it could return important results when considering different genotypes of maize plants. In this paper the following questions are addressed: (1) which machine learning models are most suitable to predict LNC and PH in maize (*Zea mays* L.) plants with spectral data from UAV-based image? and (2) amongst all predictor variables (spectral indices, bands, and the combination of both), which one is the most useful for mapping LNC and PH based on the machine learning approach?

## 2. Materials and Methods

The proposed method was divided into 4 main phases: (1) the description of in-field experiments and how the experimental design was mounted, (2) the extraction of variables LNC and PH, (3) the image preprocessing and calculation of the VIs investigated, and (4) the experimental protocol implemented. Each main phase is described in detail in the following subsections.

### 2.1. Field Trials

The experiment was carried out in the municipality of Chapadão do Sul, State of Mato Grosso do Sul, Brazil (18°46′26′′ S, 52°37′28′′ W, and an average altitude of 810 m), during the 2017/2018 and 2018/2019 crop seasons. In this experiment, 11 maize cultivars cultivated under two rates of nitrogen fertilization in topdressing, 60 kg ha$^{-1}$—considered as low and 180 kg ha$^{-1}$—considered as high, were investigated, with four replicates of each plot. The cultivars used in the experiment were: Caimbé; CatiVerde; Gorotuba; AlAvaré; BRS106; BRS4103; BRS4104; Diratininga; SCS154; SCS155; and SCS156. The dimensions of each plot were five rows, spaced at 0.45 m each, with a 5 m length. Because it corresponds to a relatively small experimental area, the soil here presents similar conditions. This area is constantly monitored and soil corrections are conducted whenever necessary.

The corn cultivars and N rates were allocated in the same plots in both seasons. The use of several cultivars and two rates of N aimed to create different situations promoted by farmers in Brazil. Thus, the models tested can estimate the variables for these conditions in both seasons. The integration between multiple varieties also was important to provide enough samples for the machine learning models to learn the necessary features: LNC and PH. It was also necessary to build a dataset heterogeneous enough to demonstrate the feasibility of these techniques. The geographic location of the area, along with the experiment plots, is displayed in Figure 1.

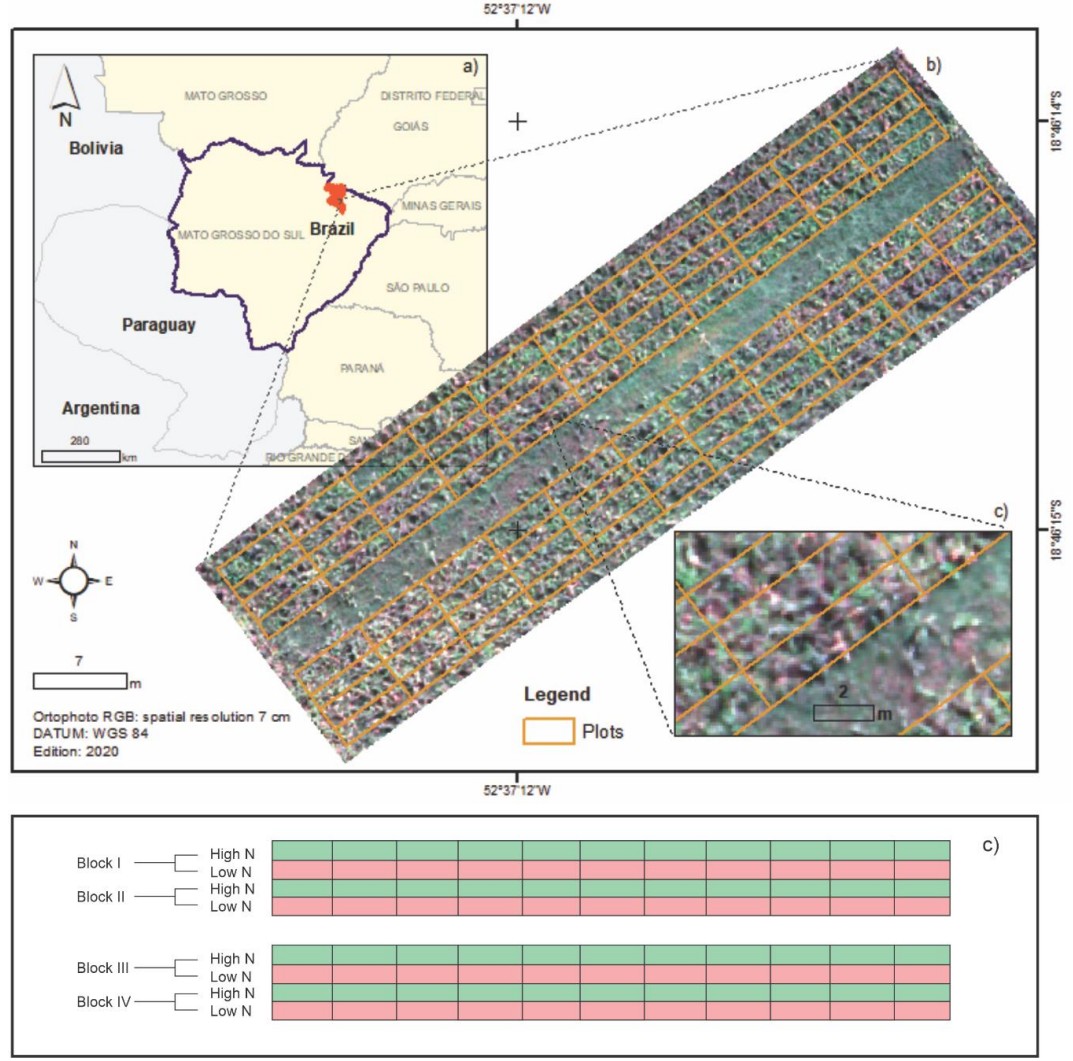

**Figure 1.** The geographic location of the study area. (**a**) Corresponds with the location of the municipality within the Mato Grosso do Sul State in Brazil; (**b**) represents the plots in the experimental site; and (**c**) is the scheme concerning the experimental fertilization rates used.

*2.2. Evaluated Variables*

Maize plants were evaluated at the V12 stage. The images were collected at this stage because plants have reached their full potential in terms of growth and nitrogen absorption in this phase. The average third of five leaves of maize plants were collected in each experimental unit. The LNC ($g\ kg^{-1}$) was obtained by the methodology described in [40]. In this regard, N was evaluated with in-field measurements and following agronomical standard procedures. For this, the Kjeldahl titration technique was applied, which is divided into (1) digestion, (2) distillation in an N distiller, and, (3) titration with sulfuric acid ($H_2SO_4$). On this same date, PH (m) was obtained with an average of five plants chosen at random in each plot. For this, a measuring tape was used, positioned from the base of the plant to its apex (i.e., the highest point of the plant; at the top of the canopy). A tracking GNSS with high precision accuracy was used to map the crop plots (yellow-grids in Figure 1), ensuring that the collected data was representative of each plot.

This provided a total of 176 in-field observations of LNC and PH. The measure mean-values of LNC and PH, for both seasons (2017/2018 and 2018/2019), did not result in statistical differences at a p-value under 0.05. For this, a Shapiro-Wilk test followed by a pairwise t student test was used. When calculating the variance for LNC, values of 21.33 g.kg$^{-1}$ and 20.21 g.kg$^{-1}$ in 2017/2018

and 2018/2019 crop seasons were obtained, respectively. As for PH, the variance obtained was 0.107 m and 0.101 m for 2017/2018 and 2018/2019 crop seasons, respectively. This information, alongside with the p-value under 0.05 for both LNC and PH in each year, indicated that the seasons returned similar conditions for both analyses.

*2.3. Image Acquisition and Vegetation Indices*

The following spectral regions were used for calculating the VIs: green (G), red (R), red-edge (RE), and near-infrared (NIR). The described wavelength (nm) is the bands' center on both sensors. The area was recorded during the first crop season (2017/2018) with a MicaSense Red-Edge multispectral sensor (G: 560 nm, R: 668 nm, RE: 717 nm, and NIR: 842 nm) embedded in a UAV-multirotor X800. For the second crop season (2018/2019), a Sensefly eBee RTK fixed-wing remotely piloted aircraft was used. The eBee was equipped with the Sensefly Parrot Sequoia multispectral sensor (G: 550 nm, R: 660 nm, RE: 735 nm, and NIR: 790 nm).

Both sensors acquired spectral data in the aforementioned wavelengths and used a luminosity sensor allowing the calibration of the acquired values. The two overflights were performed at 100 m altitude, returning a spatial image resolution (ground sample distance—GSD) of 0.10 m, and were conducted at 10:00 h (local time). Figure 2 summarizes the climatic and atmospheric conditions of each crop season (2017/2018 and 2018/2019). As described, both flights occurred at the V12 stage of each season.

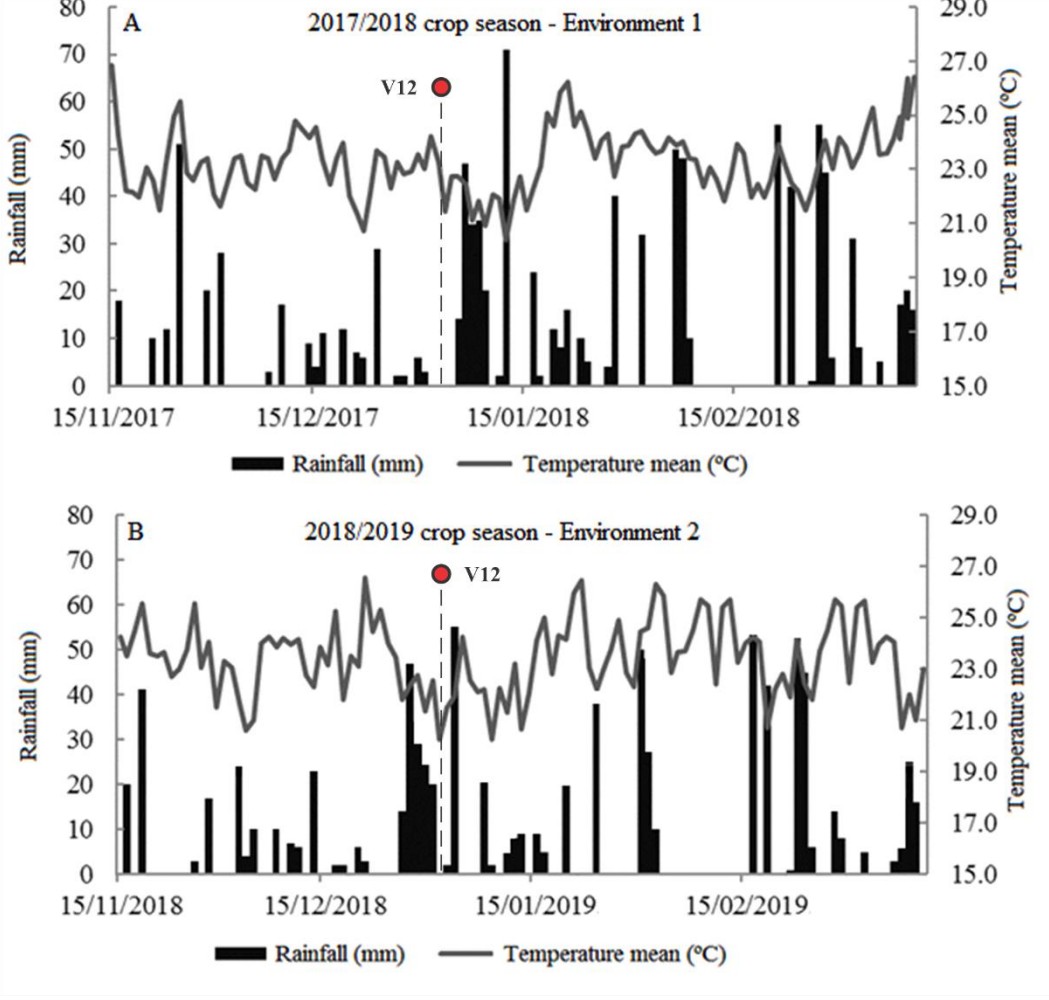

**Figure 2.** The climatic and atmospheric conditions registered during the crop seasons of 2017/2018 and 2018/2019.

For the image preprocessing, the Pix4DMapper was used, optimizing the interior and exterior parameters of the image. A sparse dense cloud based on the structure-from-motion (SfM) technique and point clouds based on the MVS (multi-view stereo) with multiple control points collected were used. These points were collected with a global navigation satellite system (GNSS), dual-frequency in real-time kinematic (RTK) mode. Images were acquired with 80% longitudinal and 60% lateral overlaps, and the digital number (DN) was converted to surface reflectance using the calibration parameters described in the manual of both sensors. The calibration and luminosity corrections were also necessary to minimize the influence of soil brightness. Because spectral indices were used in this study, this interference was also minimized. The plantation itself also was in a stage fully developed and covered most of the soil in the spatial resolution registered, making its contribution minimal to the spectral behavior of the plants.

During an experimental phase, multiple VIs were calculated with the aforementioned spectral bands. However, most of the indices did not return promising results and also presented redundancy over the tests. Because of that, only four main VIs were implemented in the machine learning models: normalized difference vegetation index (NDVI) [41], normalized difference red-edge index (NDRE) [42], green normalized difference vegetation (GNDVI) [43], and soil adjusted vegetation index (SAVI) [44]. These VIs are among the most commonly used indices to predict plant health and conditions. The equations arranged below demonstrate the spectral data used to obtain these VIs, respectively.

$$NDVI = (NIR - R)/(NIR + R) \tag{1}$$

$$NDRE = (NIR - RE)/(NIR + RE) \tag{2}$$

$$GNDVI = (NIR - G)/(NIR + G) \tag{3}$$

$$SAVI = (1 + 0.5)(NIR - R)/(NIR + R + 0.5) \tag{4}$$

## 2.4. Data Analysis

The pixel values for each plantation plot were extracted from the images. These values were used as input to estimate the measured in-field values of LNC and PH in their corresponding plot. A randomized 10-fold cross-validation sampling strategy, with a total of 10 repetitions, was used to evaluate the performance of 9 supervised machine learning models (Table 1). To evaluate the performance of each model as well as the relationship between the predicted and observed variables, the root mean squared error (RMSE) and mean absolute error (MAE) metrics were used.

The number of samples implemented was similar to others presented in previous research [12,28], which also discussed the required quantities of input data to train these types of algorithms. With the cross-validation approach, 90% of the 176 samples were used to train the models and 10% to test it. Because this process was repeated, 100 randomized test-sets were constructed. In summary, this type of validation is repeated sequentially, constantly changing the folder used for validating the algorithm [12,18,27,45]. In this manner, the algorithm is always validated with data not used at its training phase. In this experiment, the entire procedure was also repeated 100 times, which means that the models were built from scratch in every repetition.

Two decision tree-based machine learning algorithms were used here: the reduced error pruning tree with backfitting, and the random forest method with 100% of the training set as bagging size. A K-nearest neighbor was also used with three different K values: 1, 5, and 10. Support vector machines adopting sequential minimal optimization (SMO) have been tested under 2 different kernels: radial based functions and polynomial. Finally, a linear and a kernel-based regressor were also included for comparison. The linear regression uses a grid-search strategy for model selection based on the Akaike information criterion and the kernel-based regressor is a radial basis function network.

**Table 1.** Machine learning algorithms and input data variation used in this study.

| Test Order | ML Model | Reference |
|:---:|:---:|:---:|
| #1 | REPTree—REPT | Saha et al. [46] |
| #2 | Random Forest—RF | Belgiu et al. [47] |
| #3 | K-Nearest Neighbor (K=1)—1NN | Ali et al. [48] |
| #4 | K-Nearest Neighbor (K=5)—5NN | Ali et al. [48] |
| #5 | K-Nearest Neighbor (K=10)—10NN | Ali et al. [48] |
| #6 | SVM-RBF—SVMR | Nalepa et al. [49] |
| #7 | Support Vector Machine-Polynomial—SVMP | Nalepa et al. [49] |
| #8 | Linear Regression—LR | Štepanovský et al. [50] |
| #9 | RBF Regression—RBF | Cheshmberah et al. [51] |

The library default values were adopted for the number and depth of trees, nodes, and leaves in the decision tree models, as well as a different number of neighbors (1, 5, and 10) for the KNN algorithm. As stated, two functions (RBF and polynomial) were considered for SVM, the $\exp(-gamma*|u-v|^2)$ and the $(gamma*u'*v + coef0)^2$, respectively. Each value regarding the described variables was set to be calculated automatically considering the overall best predictions with an epsilon loss curve equal to 0.1. Last, a grid search approach was used to fine-tune the linear regression model (RBF Regression), thus performing a hyperparametrization of this particular model.

All the models have been tested using three sets of variables: (#1) a set with spectral-bands only (SB), (#2) a set only with Vis, and (#3) a set including both SB and VIs together. During the experimental phase, different hybrid combinations of the adopted models were evaluated. However, the combinations are not discussed in this manuscript mainly because they did not result in interesting outcomes as well as the separation between SBs and VIs. After determining the best overall algorithm, an inference model was calculated to produce a prediction map over the UAV image. This map was used to ascertain the relationship between the predicted variables and help discuss the implications of the proposal of this study.

Additionally, based on the overall best algorithm, the most contributive input data used by the learner were also identified. For this, a classifier attribute evaluation that estimates the worth of an attribute by using this specified classifier (in our case, the overall best algorithm) was implemented with the rank search as a selection method. The evaluated rank metric was based on a merit score obtained with the ZeroR regressor. This merit corresponds with the relative increase in the performance of the model in relation to the ZeroR classifier. The ZeroR was used since it takes the average value of the target variable and uses this value as a prediction. In this regard, the rank method can return a merit number even greater than 1 (since relative increase may exceed 100%). This procedure was important to determine the significance of each SB or VI to infer LNC and PH in maize crops.

## 3. Results

### 3.1. Relationship among the Agronomic Variables

To present the relationship among the evaluated variables, Figure 3 was prepared to display the Pearson's correlations between LNC and PH with the SBs and VIs evaluated in this study. The magnitude of the correlations was different for each N fertilization rate (high and low).

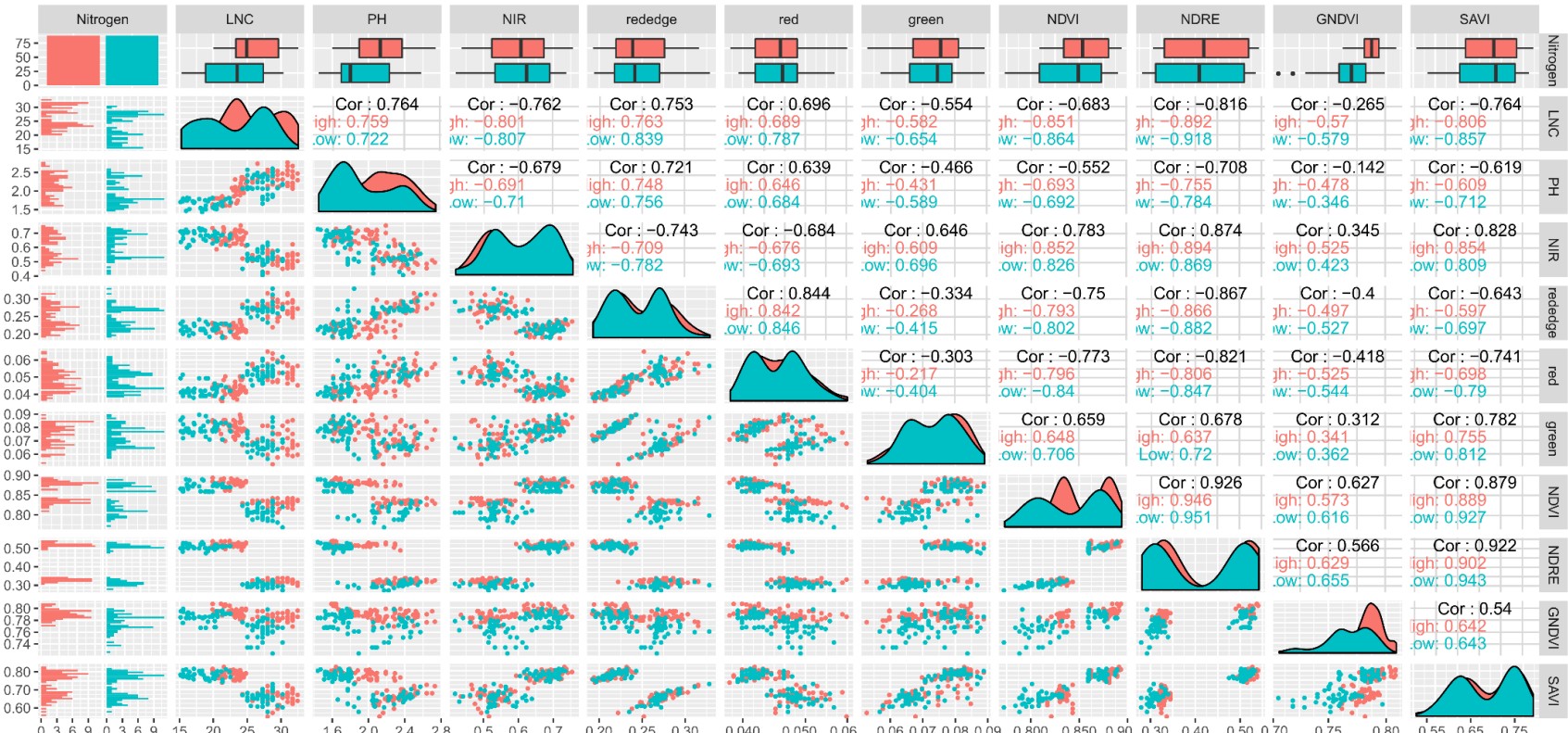

**Figure 3.** Pearson's correlation (r) between each variable implemented in this study. The lower triangle corresponds with the dispersion graphics of each correlation in a pairwise manner. The higher triangle displays the correlation value. The main diagonal illustrates a kernel density-based histogram between the two N rates and the distribution of it according to the respective variables. The top row displays box-plots of the values according to the associated N content, and the first column performs a comparison between the basic distributions of data across the two N conditions.

### 3.2. Models' Performances for LNC and PH Prediction

Figures 4 and 5 display the boxplots for the RMSE using 100 runs (10 repetitions of 10-fold cross-validation) of each machine learning algorithm under the 3 data input configurations: SB, VI, and SB+VI. Figure 4 displays the boxplot for LNC, whereas, in Figure 5, PH is displayed. Regarding the LNC estimate, the RMSE indicates a higher averaged performance of the RF model with a smaller interquartile range for the VI and SB+VI configuration. The performance of RF using only the SBs is lower than using the other configuration sets for both LNC and PH. The three KNN models also showed lower values of RMSE for VI alone than SB+VI, with a clear advantage for higher K sets (e.g., 5 and 10).

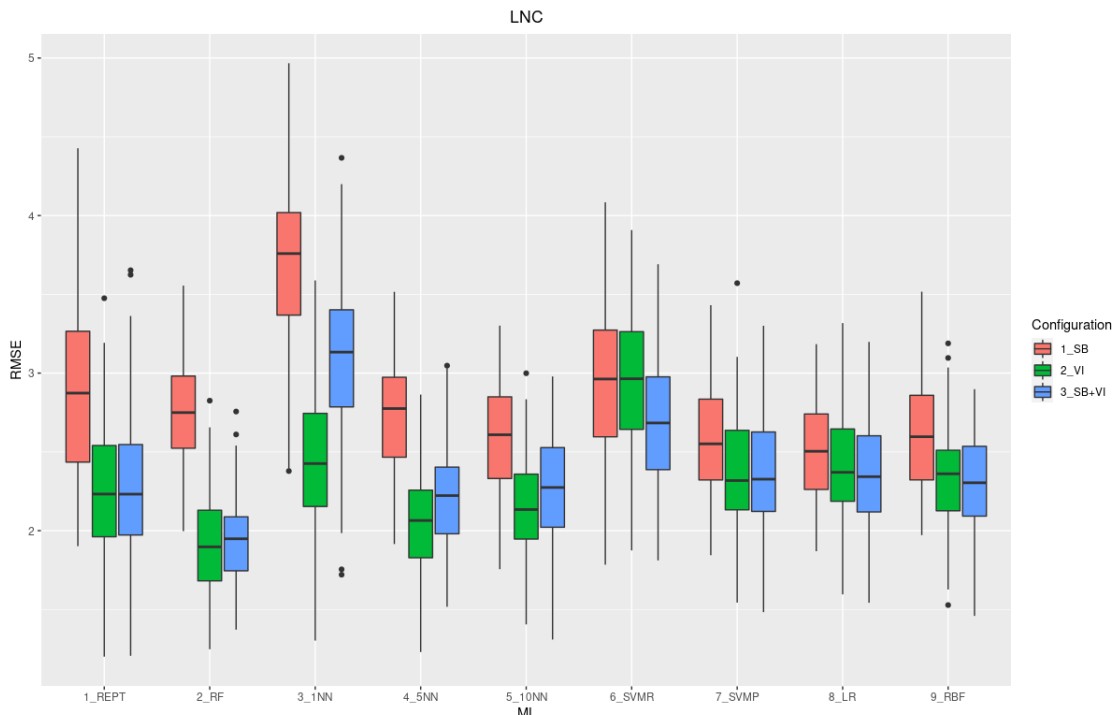

**Figure 4.** Boxplot for the root mean squared error (RMSE) for leaf nitrogen concentration (LNC) (g kg$^{-1}$) estimation using 9 machine learning algorithms, over 100 runs, and 3 input configurations: (1) spectral bands (SBs), (2) vegetation indices (VIs), and (3) SB+VI. The point plots indicate outliers encountered during the phase of the 100 different test repetitions.

In the boxplots for PH, there is a slightly lower averaged RMSE for RF when comparing it against the other models, but the combination of SBs + VIs seemed to lower the performance of the model. Beyond RF, the REPT and KNN models presented good results for the VIs dataset alone. Although some outliers were detected in the estimations, each box-plot was constructed over a 95% confidence interval. The overall performance of the best model (RF) presented an RMSE equal to 1.9 g.kg$^{-1}$ and 0.17 m, for both LNC and PH, respectively.

To better ascertain the relationship between predictions and measured variables, the regression of the overall three best methods for each variable (PH and LNC) was plotted. It used the configuration set #2 containing only the VIs as input variables (Figure 6).

The PH scatterplot in Figure 6 demonstrates how consistent the RF model was when predicting this variable. As for the LNC prediction, it is possible to notice that the two topdressing conditions of N fertilization rates are separated by the model. This is an important observation since it demonstrates that the RF approach was able to separate distinctly the low and high rate levels.

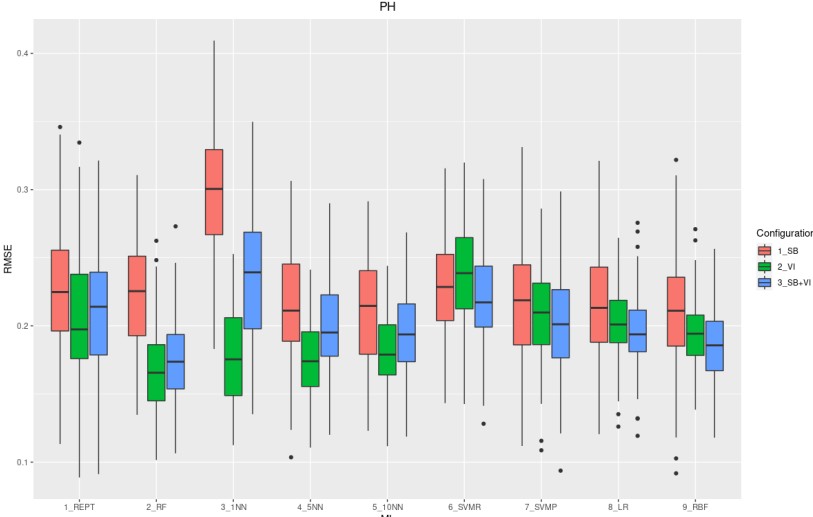

**Figure 5.** Boxplot for the root mean square error (RMSE) for plant height (PH) (m) estimation using 9 machine learning algorithms, over 100 runs, and 3 input configurations: 1) spectral bands (SBs), 2) vegetation indices (VIs), and 3) SB+VI. The point plots indicate outliers encountered during the phase of the 100 different test repetitions.

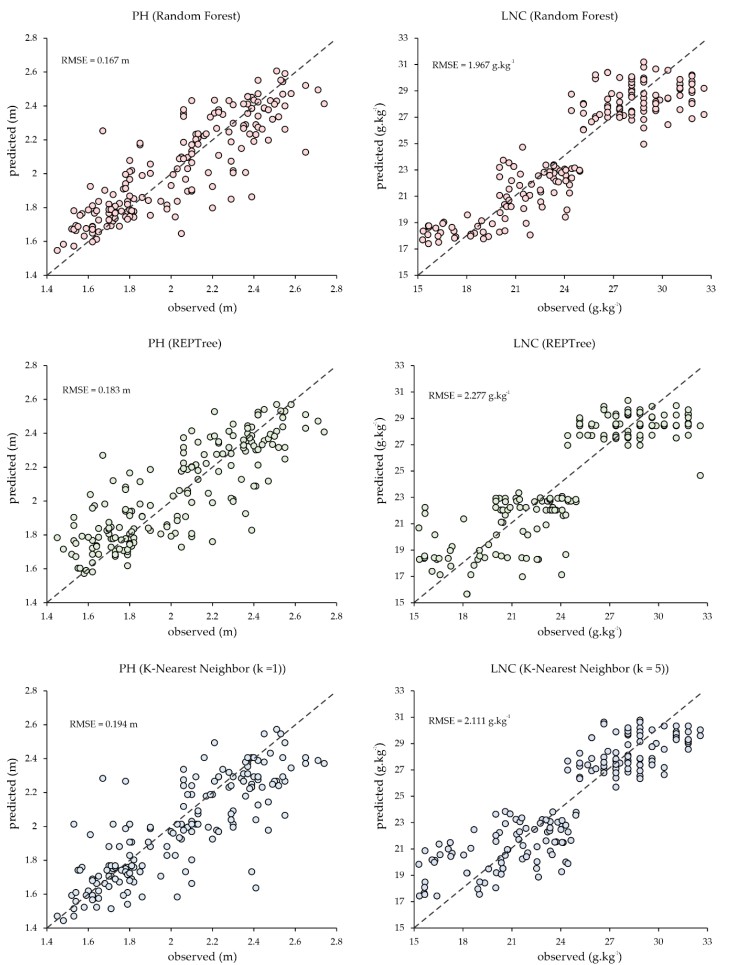

**Figure 6.** Relationship between the predicted and observed variables (LNC and PH) using the overall best models and configuration. The dashed-line corresponds with a 1:1 linear-fit.

To better demonstrate the feasibility of the proposal, a map of the predicting values for both LNC and PH was constructed with the RF model using the VIs as input parameters (Figure 7). This map can provide a qualitative approach for the result. Once trained, a machine learning model can calculate or perform inference over the image data, returning a visual representation of the area.

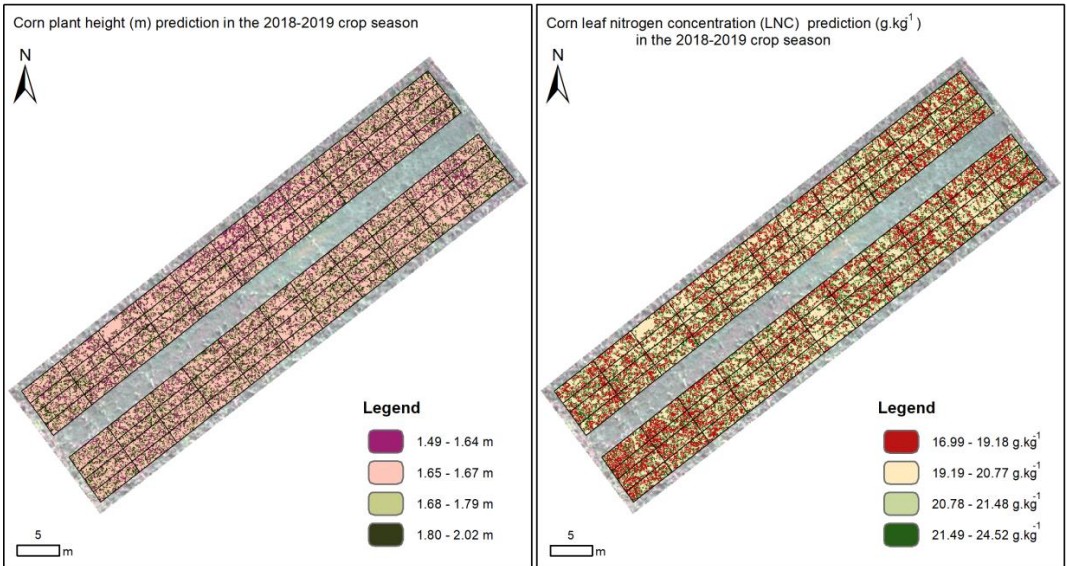

**Figure 7.** Prediction map over the 2018/2019 crop season indicating the LNC and PH related to every pixel in the unmanned aerial vehicle (UAV)-based image. The model used for inference was the random forest (RF) and only the VIs were used as input data to calculate it.

Table 2 ranks the contribution of individual attributes for RF for estimating LNC and PH. This metric was estimated with configuration #3 (VIs + SBs), considering all input data to confront both spectral bands and spectral indices' importance to the model. The merit for each attribute indicated that VIs like NDVI, NDRE, and SAVI were more contributive than SBs. This result supports the observation over previous analysis, that the VIs configuration returned better accuracy than the SBs configuration. The merit score was obtained from the ranking-based approach in the method section.

**Table 2.** Ranking order describing the merit of each machine learning model used in this study.

| Order | Attribute | Merit (avg.) - LNC | Merit (avg.) - PH |
|-------|-----------|--------------------|--------------------|
| 1 | NDVI | $1.018 \pm 0.043$ | $0.939 \pm 0.045$ |
| 2 | NDRE | $1.004 \pm 0.039$ | $0.897 \pm 0.048$ |
| 3 | SAVI | $0.912 \pm 0.047$ | $0.862 \pm 0.046$ |
| 4 | Red-Edge | $0.88 \pm 0.049$ | $0.802 \pm 0.046$ |
| 5 | Near-Infrared | $0.842 \pm 0.04$ | $0.734 \pm 0.046$ |
| 6 | Red | $0.828 \pm 0.047$ | $0.719 \pm 0.043$ |
| 7 | Green | $0.714 \pm 0.052$ | $0.596 \pm 0.046$ |
| 8 | GNDVI | $0.454 \pm 0.044$ | $0.499 \pm 0.045$ |

## 4. Discussion

The evaluation of multiple cultivars and different quantities of N fertilizer was implemented to simulate the characteristics encountered in most maize-crops around Brazil. With this experimental design, using the spectral data in three distinct configurations, we investigated the performance of a set of machine learning algorithms, like the REPT, RF, KNN, SVM (with RBF and polynomial kernels), and LR. The leaners returned similar predictions according to the respective configuration set when predicting both LNC and PH. The adopted configurations were indicative of the importance of VIs in the prediction of these agronomic variables. In a direct analysis regarding the relationship between

each variable (Figure 3), a negative correlation of LNC with NIR and VIs and a positive correlation with R was found. This observation differentiates from the literature [10–17] since the LNC is closely related to chlorophyll influence over these spectral variables. Still, this could configure a particular case related to field conditions raised in this study. The same observations were also noted in a previous study, conducted in a Citrus orchard, which also observed the same spectral bands from the Parrot Sequoia embedded sensor [11].

The RF algorithm may share a similar trend in the nutrient analysis since different and recent types of research concluded that this learner is obtaining optimal and balanced results with different spectral data. In similar research [52], an experiment conducted with maize and multispectral imagery from the orbital scale demonstrated that VIs showed strong performance. Another research [34], aiming to estimate maize, stated that the RF learner returned the highest accuracies among the evaluated algorithms. For N content, although not conducted in maize crops, multiple types of research [25,33,53–56] also concluded that the RF learner, as well as other types of regressors based on decision trees, were appropriate to model LNC. In the presented approach, the errors encountered with this model are relatively lower or similar when in comparison to the aforementioned studies.

RF is one of the most powerful methods in the current literature related to machine learning tasks [57–62]. The increase in data dimensionality is often seen as a problem for most traditional methods. In this study, the increase in dimensionality was also necessary to improve the overall accuracy of this model. As for limitations, the major difficulty associated with this method, as well as the other machine learning approaches, is the small amount of data [63]. However, as agronomic variables are onerous to obtain, tests with multiple repetitions and configurations sets were conducted to ensure the accuracy of this proposal. The strategy of adopting different configurations and repetitions should be further explored in future research, where the number of instances is relatively small.

The performance of each algorithm was, as discussed, evaluated with different configurations. This analysis returned interesting outcomes, as the accuracy of the learners were better with the VIs as attributes (Figures 4 and 5). The importance of VIs as estimators of N and PH was evaluated in previous papers [18,52,64–67]. This is mainly because the VIs enhance some characteristics related to biological variables, such as chlorophyll content and biomass, which are highly correlated with LNC and PH. Nonetheless, when implemented in machine learning methods, it is difficult to understand the exact function in the model's predictions. However, when considering different scenarios, as well as implementing the rank-based approach presented here, it is possible to shine some light onto this process. The rank demonstrated that most contributions are provided by the VIs, and, to a lesser extent, the SBs with their respective surface reflectance values. This type of evaluation is important since it provides a matter to indicate which input variables are more suitable to model the evaluated problem, which can reduce the amount of data input, resulting in an accurate and more rapid estimative.

As for the image itself, the major limitation of a UAV image data collection is the low capacity to compensate and analyze larger areas. However, this type of aerial remote sensing is important when considering the spatial resolution and highly detailed information obtained on the vegetation cover, permitting an analysis at a plant or crop-plot level [68–71]. Additionally, by evaluating crop at an aerial view, it is easier to ascertain the relationship between spectral data and biophysical variables, since the end-user can reduce the amount of noise introduced in the system by extracting only pixels corresponding with the canopy itself.

The approach presented here may also be implemented with different datasets over diverse areas, crops, and sensors. The approach of adopting multiple machine learning models and VIs could also be used to predict agronomic variables like other macronutrients and micronutrients. Previous experiments already suggest the possibility of inference other nutrients with spectral data from proximal sensors [12,62,72]. In this regard, additional experiments could consider multispectral data from sensors embedded in UAVs. Here, the particular objective was to investigate the contribution of multispectral data in machine learning methods to nutrient content (N) and height (PH). The advantage of LNC and PH prediction with UAV-based images is that it promotes a rapid and cost-efficient manner

to the recurrent monitoring of the agricultural landscapes. However, the traditional agronomic method should not be substituted but assisted by remote sensing technologies and computational techniques such as the ones indicated here.

## 5. Conclusions

In this study, a machine learning approach was implemented to estimate LNC (g kg$^{-1}$) and PH (m) for maize plants. It was tested whether the models are impacted by data input regarding different combinations of SBs and VIs. It also demonstrated which one of the implemented learners is more suitable to predict both parameters (LNC and PH). The conducted experiment showed that the RF algorithm performed better, with RMSE equal to 1.9 g.kg$^{-1}$ and 0.17 m, for LNC and PH, respectively. The VIs contributed more to the algorithm's performances than the SBs. This paper concludes that the proposed approach of machine learning models is appropriate to predict these agronomic variables. This method may be used in research that intends to evaluate different types of crops or applied in precision agriculture practices and assist in decision-making models. Regardless, future experiments should be conducted in more practical conditions.

**Author Contributions:** Conceptualization, H.P., J.M.J., and Y.Y.; methodology, P.E.T., F.H.R.B., C.A.d.S.J., H.P., J.M.J.; investigation, H.P.; writing—original draft preparation, H.P., L.P.O, J.M.J. A.P.M.R., D.E.G.F.; writing—review and editing, L.P.O., J.M.J., A.P.M.R., W.N.G.; project administration, P.E.T., F.H.R.B., L.P.R.T., D.C.S. All authors have read and agreed to the published version of the manuscript.

**Funding:** This research was funded by CNPq, grant number 303559/2019-5, 433783/2018-4, 314902/2018-0, and 304173/2016-9; CAPES, grant number 88881.311850/2018-01, and; Fundect, grand number 59/300.066/2015, and 59/300.095/2015.

**Acknowledgments:** The authors acknowledge the support of UFMS (Federal University of Mato Grosso do Sul), UCDB (Dom Bosco Catholic University), CNPq (National Council for Scientific and Technological), and CAPES (Coordination for the Improvement of Higher Education Personnel - Finance code 001).

**Conflicts of Interest:** The authors declare no conflict of interest. The funders had no role in the design of the study; in the collection, analyses, or interpretation of data; in the writing of the manuscript, or in the decision to publish the results.

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
