# Peer review of "Leaf Nitrogen Concentration and Plant Height Prediction for Maize Using UAV-Based Multispectral Imagery and Machine Learning Techniques"

_remotesensing, doi:10.3390/rs12193237_

Round 1
Reviewer 1 Report
My major issue still concerns the comparison between observed agronomic variables (LNC and PH) and the predicted ones. Adding a new figure with dispersion plots does not solve this issue.
It is wrong to perform a linear regression between these as shown in figure 6 (red line). The only correct way is to compare observed and predicted values by analyzing their difference, so by comparing them with the 1:1 line (dashed line). In this respect, one may use the RMSE or MAE (both based on the direct comparison between observed and predicted values) for evaluating the results. Calculation of r or R2 after another linear regression is not allowed.
I cannot approve the manuscript as long as this mistake is in the results section.
In addition, I noted a few minor errors in the text:
Line 14: ‘g.kg1 ’ should be ‘g.kg-1 ’
Line 135: ‘flight’ should be ‘recorded’
Line 299: ‘N fertilized’ should be ‘N fertilizer’
Line 301: ‘in the three distinct configurations’ should be ‘in three distinct configurations’
Author Response
Dear reviewer,
We agreed with your concern. We removed the R², the regression line, and the equation from figure 6 and used RMSE to compare the difference between observed and predicted values. We also corrected the minor errors in the text. We hope that you find these alterations appropriate. Thank you.
Reviewer 2 Report
Publish as is
Author Response
Thank you for taking the time to review our manuscript.
Kind regards,
Authors.
Reviewer 3 Report
The subject addressed in the manuscript is of interest to remote sensing, and the application of machine learning is a very up to date topic. However, the text needs improvement, as there are structural flaws that need to be corrected for publication in a high-impact international journal. The main problem of the work is in the discussion of the results. Below I specify the questions I encountered throughout the text.
INTRODUCTION: this part of the text needs to be synthesized; it must be more cohesive and specific to the problem proposed to study.
Lines 22-27: I don't see any sense in this paragraph; it doesn't match the quality of the desired journal. In some parts of the text, it looks like a dissertation or thesis. Be careful with that.
Lines 57-58: Why is machine learning still a relevant topic? Clarify and complement.
Line 68: What do you mean by "production"?
Lines 71-76: This part shows the bias that the work presents when using RF. I suggest you judge whether such studies citation makes sense in a well-constructed introduction. Often, the mere citation of previous works should not be made. The main ideas must be linked to the research goal.
MATERIALS AND METHODS:
Line 103: Were the corn cultivars and N rates allocated in the same plots in both seasons? If so, the effect of the lack of N may have maximized the response in terms of vegetation vigor. Such issue highlights the concern that I pointed out further in the Conclusion item about how these results could be expanded to real field conditions.
Line 116: Why were the images collected on the V12 stage?
Line 116: change "for corn crop" by "of corn plants".
Line 122: what do you mean by "apex"? Top of the canopy? Leaf? Explain.
Line 131: is just enough variance to state that both years were in the same condition?
Line 138: "with autonomous control of takeoff, flight plan, and landing" - this information is unnecessary.
Lines 133-142: why did you employ different sensors? Could this influence the results?
Lines 166-167: was any image segmentation done to remove the soil reflectance? Why?
Lines 170-171: Why were so similar statistics (R to R2; RMSE to MAE) used? How do they complement each other?
Table 1 and lines 184-191: It was not clear what the acronyms mean; moreover, there is no relation to the acronyms used in the figures in the results item.
Lines 192-193: "run over an Intel® Core ™ i7 CPU and 12Gb RAM workstation" - is this a relevant information? Would the results be different if you have used an AMD or i5, 8Gb or 32 Gb RAM?
Lines 202-203: "Boxplots for all these configurations were also produced" - Remove such information. You do not have to inform this kind of issue. Pay attention to the scientific writing style.
RESULTS:
Lines 219-230: How does this paragraph assist in the discussion and conclusion of the research?
Lines 238-241: This has already been explained in the figure title. It should not be placed in the text again. Correct throughout the text.
Line 254: an index must accompany acronyms. The reader should not have to return to the methodology topic to understand what they are.
Lines 263-264: "It used the configuration set # 2 containing" - where is this numbering?
Line 265-266: Why did the comparison metrics change here?
Figure 6: Be more specific about the content of the figure in its title.
Lines 273-274: What should the reader interpret these graphs? What is the primary information?
Lines 281-282: Other fields? Based on what result could you stated this? Conditions can change completely one another season or field.
Figure 8: If the soil reflectance was not removed by segmentation, the height and N content estimates are wrong, since there is no zero value, right? Is there soil in the image? It would help if you discussed this.
Line 291: Why was this configuration adopted for this analysis?
Lines 294-295: explain how the reader can interpret this result.
DISCUSSION:
Line 306: "influenced by the water in the structure" - water in which structure? That is not what influences the NIR reflectance.
Lines 306-307: "similar results in G" - similar to what?
Lines 304-311: Dispensable paragraph. All this knowledge about spectral behavior is widespread.
Lines 315-316: Discuss more this issue of the change in the direction of the correlation. What's new about this? What is its practical implication?
Line 324: "was more appropriate" - more appropriate than what?
Lines 330-332: Where are such additional tests? Clarify.
Line 332: "This strategy" - which strategy are you referring to?
Lines 334-336: If VIs have the spectral bands as their primary constituents, why using individual bands did not generate equivalent results? This is the kind of discussion that needs to be done, explaining to the reader why this occurs.
Lines 338-340: explain this question to the reader.
Lines 341-347: this is a result already presented; there is no discussion here.
Lines 347-348: "important" for whom or what? From my perspective, the authors did not explain why such analysis would be useful.
Lines 350-362: Paragraphs without relevance and with missing arguments. As it stands, it is characterized more as a research justification than as a discussion of results.
Lines 362-364: this is a result already presented; there is no discussion here.
Lines 366-367: how could it be predicted? When doing SR survey, the main inference is about vegetation vigor. How would it be possible to isolate the effect of nutrients to model them and achieve proper accuracy in a real field situation?
CONCLUSIONS: There is a summary of your results on this topic. Here you must answer to the initial study question.
Line 381: "performance than other attributes" - which attributes? The issue of individual bands has already been said in the previous sentence.
Lines 381-383: Is the model effective to estimate the N content in the leaves? In the graphs, I see two large groups that govern the macro behavior of the estimates. In the field situation, where there is no such variability forced by the application of N, how would the quality of the predictions be? I believe it would be much lower. Such issue needs to be discussed in the text. I believe that this is the main problem and/or limitation of the manuscript.
Author Response
Dear reviewer,
Thank you for taking the time to evaluate our manuscript. We address, point-by-point, each concern expressed during your revision. We hope that you find the alterations appropriate. We also would like to state that your revision helped us improve the overall quality of our manuscript.
INTRODUCTION: this part of the text needs to be synthesized; it must be more cohesive and specific to the problem proposed to study.
Lines 22-27: I don't see any sense in this paragraph; it doesn't match the quality of the desired journal. In some parts of the text, it looks like a dissertation or thesis. Be careful with that.
R: We agree with this observation and removed the paragraph in the introduction section.
Lines 57-58: Why is machine learning still a relevant topic? Clarify and complement.
R: We added the following sentence to complement this phrase: “However, performing this task with machine learning techniques is still a recent and relevant topic in agriculture remote sensing since it provides a robust and direct approach to evaluate different agronomic variables.”
Line 68: What do you mean by "production"?
R: We meant “yield”. We changed the text accordingly.
Lines 71-76: This part shows the bias that the work presents when using RF. I suggest you judge whether such studies citation makes sense in a well-constructed introduction. Often, the mere citation of previous works should not be made. The main ideas must be linked to the research goal.
R: We agree with this observation and, although the original intention was to establish a link with the overall performance of the RF in our study, we comprehend how this affects our introduction. In this regard, we remove it and modify the content, as in: “The combination of machine learning techniques and vegetation indices (VI)s is also an important subject in agricultural applications and been adopted in different studies, some of which related to corn characteristics predictions [34,35].”
MATERIALS AND METHODS:
Line 103: Were the corn cultivars and N rates allocated in the same plots in both seasons? If so, the effect of the lack of N may have maximized the response in terms of vegetation vigor. Such issue highlights the concern that I pointed out further in the Conclusion item about how these results could be expanded to real field conditions.
R: They were the same in both seasons. Regarding the concern over the conclusion, we addressed it accordingly in later responses.
Line 116: Why were the images collected on the V12 stage?
R: The images were collected on the V12 stage because plants have reached their full potential in terms of growth and nitrogen absorption in this phase. We added this information alongside the observed phrase.
Line 116: change "for corn crop" by "of corn plants".
R: Done. Thank you.
Line 122: what do you mean by "apex"? Top of the canopy? Leaf? Explain.
R: It corresponds with the highest point of the plant; at the top of the canopy. We added this information alongside this phrase.
Line 131: is just enough variance to state that both years were in the same condition?
R: Since no statistical difference (p-value < 0.05) was observed between each variable in each year, we inserted this information alongside with the variance to help ascertain that they returned similar results in both seasons.
Line 138: "with autonomous control of takeoff, flight plan, and landing" - this information is unnecessary.
R: This information was removed.
Lines 133-142: why did you employ different sensors? Could this influence the results?
R: We used different sensors simply because of its availability for each year, and because we wanted to observe if the different sensors produced different information from the remote sensing point of view. For spectral variation, this was not the case, as both MicaSense and Sequoia are related to each other and operate at the same spectral bands.
Lines 166-167: was any image segmentation done to remove the soil reflectance? Why?
R: No image segmentation was conducted for this purpose. We performed the calibration and luminosity corrections necessary to minimize the influence of soil brightness during the pre-processing stage of the images. Since we used spectral indices, this interference was also minimized. The plantation itself also was in a stage fully developed, and covered most of the soil, being almost impossible to segment it in the spatial resolution registered. We, however, are confident that the contribution from the soil was minimal in this dataset. We added this information to help the reader to better comprehend these conditions. Thank you.
Lines 170-171: Why were so similar statistics (R to R2; RMSE to MAE) used? How do they complement each other?
R: Indeed, we agreed with the redundancy between these statistics. Because of that, we removed some of them and used only RMSE and MAE. R was only used for the comparison between different agronomic variables.
Table 1 and lines 184-191: It was not clear what the acronyms mean; moreover, there is no relation to the acronyms used in the figures in the results item.
R: We removed the acronyms from the text lines since it caused some confusion. The ones used in Figures 4 and 5 are related to the ones on the table, with the order in which each test was conducted. We changed them in Figure 6 and rewrite the ones in the discussion section. Thank you for this observation.
Lines 192-193: "run over an Intel® Core ™ i7 CPU and 12Gb RAM workstation" - is this a relevant information? Would the results be different if you have used an AMD or i5, 8Gb or 32 Gb RAM?
R: We agree with this observation and removed this information. The results were not varied because of that; only the processing time, which we did not present because the training and testing phase returned similar CPU time for all the models.
Lines 202-203: "Boxplots for all these configurations were also produced" - Remove such information. You do not have to inform this kind of issue. Pay attention to the scientific writing style.
R: Done. Thank you.
RESULTS:
Lines 219-230: How does this paragraph assist in the discussion and conclusion of the research?
R: We inserted this Figure to better ascertain the relationship between each agronomic variable before being evaluated by the machine learning models. Based on the next observation, we remove most of the text and leave only the Figure presentation.
Lines 238-241: This has already been explained in the figure title. It should not be placed in the text again. Correct throughout the text.
R: We agreed that this information is redundant and choose to remove the previous text.
Line 254: an index must accompany acronyms. The reader should not have to return to the methodology topic to understand what they are.
R: We added them to the Figures legend.
Lines 263-264: "It used the configuration set # 2 containing" - where is this numbering?
R: Indeed, we mistakenly used A, B, and C in the Method section. We now corrected it by using the same numbering system used in the rest of the manuscript:
Lines 192-193: “All the models have been tested using three sets of variables: (#1) a set with spectral-bands only (SB), (#2) a set only with VIs, and, (#3) a set including both SB and VIs together.”
Line 265-266: Why did the comparison metrics change here?
R: We removed this phrase since we removed the redundant results using R². We also changed the Figure 6 because of it.
Figure 6: Be more specific about the content of the figure in its title.
R: We altered the figure title. It now reads: “Figure 6. Relationship between the predicted and observed variables (LNC and PH) using the overall best models and configuration. The dashed-line corresponds with a 1:1 linear-fit.”
Lines 273-274: What should the reader interpret these graphs? What is the primary information?
R: We removed these lines as well the Figure 7 because it was presenting redundant information.
Lines 281-282: Other fields? Based on what result could you stated this? Conditions can change completely one another season or field.
R: The original intent of this affirmative was to state that the RF leaner, with the same configuration of spectral indices, can be applied in other areas, as long as it is trained with data from there. We now can see how this is somewhat confusing and incomplete. To avoid confusion, we choose to remove the phrase.
Figure 8: If the soil reflectance was not removed by segmentation, the height and N content estimates are wrong, since there is no zero value, right? Is there soil in the image? It would help if you discussed this.
R: Actually, the prediction on PH did not return zero mainly because, in this spatial resolution, all of the pixels were composed of plants. An evaluation of our digital surface model generated during the pre-processing stage of the orthomosaic also corroborated this observation mainly because no lower or zero values were encountered in the plantation plots.
Line 291: Why was this configuration adopted for this analysis?
R: This set was used to demonstrate the individual contribution of all the input variables, being a spectral band or a spectral index.
Lines 294-295: explain how the reader can interpret this result.
R: In the method section (lines 204-205) we stated that “This Merit corresponds with the relative increase in the performance of the model in relation to the ZeroR classifier.” In this regard, we demonstrated the relative increase (for example, 1.018 is equal to 101.8%) in relation to the results returned by a base learner, ZeroR. In short, we use the ZeroR leaner to establish a base for the evaluated metrics, and the metric score is important to understand this difference.
DISCUSSION:
Line 306: "influenced by the water in the structure" - water in which structure? That is not what influences the NIR reflectance.
R: We removed this information.
Lines 306-307: "similar results in G" - similar to what?
R: We removed this information.
Lines 304-311: Dispensable paragraph. All this knowledge about spectral behavior is widespread.
R: We removed this paragraph. Thank you for pointing this out.
Lines 315-316: Discuss more this issue of the change in the direction of the correlation. What's new about this? What is its practical implication?
R: Indeed, this information is of little practical implication, and only configures an observation encountered in our study. Because of that, we modified this paragraph and inserted a new statement based on the contribution of another reviewer. The new paragraph reads as follows: “In a direct analysis regarding the relationship between each variable (Figure 3), a negative correlation of LNC with NIR and VIs and a positive correlation with R was found. This observation differentiates from the literature [10–52] since the LNC is closely related to chlorophyll influence over these spectral variables. Still, this could configure a particular case related to field conditions raised in this study. The same observations were also noted in a previous study, conducted in a Citrus orchard, which also observed the same spectral bands from the Parrot Sequoia embedded sensor [11].”
Line 324: "was more appropriate" - more appropriate than what?
R: We exchanged the line for: “For N content, although not conducted in corn crops, multiple types of research [25,33,53–56] also concluded that the RF learner, as well as other types of regressors based on decision trees, were appropriate to model LNC.”
Lines 330-332: Where are such additional tests? Clarify.
R: We removed the word “additional” since we referred to the tests already conducted in our research. Sorry for this mistake.
Line 332: "This strategy" - which strategy are you referring to?
R: We referred to the multiple configurations and repetitions strategy. The phrase now reads as: “The strategy of adopting different configurations and repetitions should be further explored in future research, where the number of instances is relatively small.”
Lines 334-336: If VIs have the spectral bands as their primary constituents, why using individual bands did not generate equivalent results? This is the kind of discussion that needs to be done, explaining to the reader why this occurs.
R: Its mainly because the VIs enhances characteristics related to biological variables. The ones used in this study are specifically related to chlorophyll content and biomass, which are highly correlated with LNC and PH. We have added this information to the discussion section.
Lines 338-340: explain this question to the reader.
R: Based upon the next observation, we remove part of the information presented in the next paragraph and connected with the lines presented here. We also explained the importance of the rank approach in the same paragraph.
The new paragraph reads as: “The performance of each algorithm was, as discussed, evaluated with different configurations. This analysis returned interesting outcomes, as the accuracy of the learners were better with the VIs as attributes (Figures 4 and 5). The importance of VIs as estimators of N and PH was evaluated in previous papers [18,52,64–67]. Nonetheless, when implemented in machine learning methods, it is difficult to understand the exact function in the model’s predictions. However, when considering different scenarios, as well as implementing the rank-based approach presented here, it is possible to shine some light into this process. The rank demonstrated that most contributions are provided by the VIs, and, to a lesser extent, the SBs with their respective surface reflectance values. This type of evaluation is important since it provides a matter to indicate which input variables are more suitable to model the evaluated problem, which can reduce the amount of data input, resulting in an accurate and more rapid estimative.”
Lines 341-347: this is a result already presented; there is no discussion here.
R: The reason was explained in the previous suggestion.
Lines 347-348: "important" for whom or what? From my perspective, the authors did not explain why such analysis would be useful.
R: We have added this information to the concluding phrases of the new paragraph. The importance of identifying the best input variables into the model is that we can indicate what indices or bands present a better contribution to the algorithm, thus reducing the processing time needed to prepare a complex dataset, and maybe reduce the computational cost of evaluating “more than enough” input data.
Lines 350-362: Paragraphs without relevance and with missing arguments. As it stands, it is characterized more as a research justification than as a discussion of results.
R: We agreed and removed the paragraph.
Lines 362-364: this is a result already presented; there is no discussion here.
R: We removed this information.
Lines 366-367: how could it be predicted? When doing SR survey, the main inference is about vegetation vigor. How would it be possible to isolate the effect of nutrients to model them and achieve proper accuracy in a real field situation?
R: We performed some other experiments and were able to determine some relation between nutrients and spectral data provided by proximal sensors. Indeed, there is a gap in the literature when it comes to UAV-based images and multispectral bands. To avoid such confusion, we have modified this phrase and inserted references to support this information. The new phrase reads as follows: The approach of adopting multiple machine learning models and VIs could also be used to predict agronomic variables like other macronutrients and micronutrients. Previous experiments already suggest the possibility of inference other nutrients with spectral data from proximal sensors [12,62,72]. In this regard, additional experiments could consider multispectral data from sensors embedded in UAVs.
CONCLUSIONS: There is a summary of your results on this topic. Here you must answer to the initial study question.
Line 381: "performance than other attributes" - which attributes? The issue of individual bands has already been said in the previous sentence.
R: We agree that this information is redundant and choose to remove the phrase.
Lines 381-383: Is the model effective to estimate the N content in the leaves? In the graphs, I see two large groups that govern the macro behavior of the estimates. In the field situation, where there is no such variability forced by the application of N, how would the quality of the predictions be? I believe it would be much lower. Such issue needs to be discussed in the text. I believe that this is the main problem and/or limitation of the manuscript.
R: Indeed, the inference model is limited to the conditions observed in the experimental field. Because of that, future experiments should be conducted in more natural/practical conditions. We inserted this limitation in the concluding remarks.
Reviewer 4 Report
The manuscript deals with the retrieval of leaf nitrogen concentration in maize considering also plant height in UAV imagery exploiting machine learning techniques, it presents good results and level of innovation in the field, considering also that it makes use of multispectral imagery solely.
The manuscript is clear, well-written and structured, the experimental plan is well-designed and results are consistent.
I have minor comments and a concern about the discussion of results.
Minor comments:
- Usually in agronomy corn is mentioned as maize (like in paper n. 17);
- The 11 cultivars are several but are not listed;
- It is clear, also thanks to Figure 2, that the two crop seasons are very similar, but what about soil conditions? Could they be considered sufficiently homogeneous over the study area, so that variability in biomass is not related to this limiting factor? I guess it is like that, so please state it;
- page 9 lines 240-241 the figure reference numbers are wrong;
- In Figure 8 spatial variability is not so evident, especially regarding PH in the first panel, maybe a different colour ramp and categorisation could help enhancing variability at the plot scale, also considering the very high resolution of UAV imagery. Maybe an indication of low and high fertilizer rates in the different experimental plots in Figure 1 could help the general understanding.
Important note (major concern):
In conclusions there is the statement: “The reflectance values in the R band are directly linked (positive correlation) with the LNC, corroborating the structure (photosynthetic pigments and plant cell structure).” This is not true by a plant physiology point of view: the photosynthetic process absorbs light in the red region, so the higher the chlorophyll content the less the reflectance in the red band (this is one basic principle of VIs).
This issue is also connected with the discussion of results: surprisingly enough you found negative correlation of LNC with NIR and VIs, which is not in line with the literature, but it could represent a particular case (given you have not a time series, but just one image at a given growing stage): this could actually happen depending on the dilution of N during biomass growth. Small plants could have higher LNC than plant more developed ones. But in this case it could not happen that negative correlation is also found for PH. Since your experimental plan is sounding and your data collection and processing seems reliable, I think you have to find an explanation (different from the one cited above), and possibly some references in the literature, for that.
Author Response
Dear reviewer,
Thank you very much for providing this feedback. We are pretty content with all the observations and remarks presented in your revision and agreed with all of them. We believe that they addressed issues that are important to be detailed and corrected. We think your revision improved the overall quality of our manuscript. For that, again, we thank you. We prepared a point-by-point response to your statements. Please see below.
Usually in agronomy corn is mentioned as maize (like in paper n. 17);
R: We agreed and exchanged the term to maize.
The 11 cultivars are several but are not listed;
R: We listed all of them in the method section, as it reads in: “The cultivars used in the experiment were: Caimbé; CatiVerde; Gorotuba; AlAvaré; BRS106; BRS4103; BRS4104; Diratininga; SCS154; SCS155 and; SCS156.”
It is clear, also thanks to Figure 2, that the two crop seasons are very similar, but what about soil conditions? Could they be considered sufficiently homogeneous over the study area, so that variability in biomass is not related to this limiting factor? I guess it is like that, so please state it;
R: Indeed, since it corresponds to a small experimental area, the soil presents similar conditions. This area is constantly monitored and corrections are conducted whenever necessary. Regardless, thank you for this question. We added this information as well in the new version to avoid confusion.
page 9 lines 240-241 the figure reference numbers are wrong;
R: We corrected it.
In Figure 8 spatial variability is not so evident, especially regarding PH in the first panel, maybe a different colour ramp and categorisation could help enhancing variability at the plot scale, also considering the very high resolution of UAV imagery. Maybe an indication of low and high fertilizer rates in the different experimental plots in Figure 1 could help the general understanding.
R: We agree with this observation. In this regard, we have inserted in Figure 1 the scheme concerning the experimental fertilization rates. For Figure 8, we traded the color ramp to better enhance the visualization. We also increased the transparency from the UAV imagery in the background and reduced the number of classes (from 5 to 4) to minimize the amount of visual pollution. Thank you.
In conclusions there is the statement: “The reflectance values in the R band are directly linked (positive correlation) with the LNC, corroborating the structure (photosynthetic pigments and plant cell structure).” This is not true by a plant physiology point of view: the photosynthetic process absorbs light in the red region, so the higher the chlorophyll content the less the reflectance in the red band (this is one basic principle of VIs).
This issue is also connected with the discussion of results: surprisingly enough you found negative correlation of LNC with NIR and VIs, which is not in line with the literature, but it could represent a particular case (given you have not a time series, but just one image at a given growing stage): this could actually happen depending on the dilution of N during biomass growth. Small plants could have higher LNC than plant more developed ones. But in this case it could not happen that negative correlation is also found for PH. Since your experimental plan is sounding and your data collection and processing seems reliable, I think you have to find an explanation (different from the one cited above), and possibly some references in the literature, for that.
R: We agreed with this observation and, also based upon another revision, we choose to remove part of this statement. However, we inserted a discussion regarding it. We also agreed that this information is not being in line with the current literature. Still, this not only occurred in this study but in another similar study of ours with Citrus plants that used a Parrot Sequoia sensor, like the one used here. We mention this issue in the following statement: “In a direct analysis regarding the relationship between each variable (Figure 3), a negative correlation of LNC with NIR and VIs and a positive correlation with R was found. This observation differentiates from the literature [10–17] since the LNC is closely related to chlorophyll influence over these spectral variables. Still, this could configure a particular case related to field conditions raised in this study. The same observations were also noted in a previous study, conducted in a Citrus orchard, which also observed the same spectral bands from the Parrot Sequoia embedded sensor [11].”
Again, thank you very much for your contribution. We hope that you find the alterations appropriate.
Kind regards,
Authors.
Round 2
Reviewer 3 Report
I am satisfied with the changes made and the answers provided by the authors to the questions raised by me. Thus, for my part, I consider that the manuscript can follow for publication.
Author Response
Thank you kindly for your valuable revision.
This manuscript is a resubmission of an earlier submission. The following is a list of the peer review reports and author responses from that submission.
Round 1
Reviewer 1 Report
It's very interesting manuscript, especially due to UAV usage. But at the same time authors can improve result's description. Namely,
- Too short description of the in-situ data for model training and validation, and data collection procedure itself. How data was collected? What is the dataset size?
- What is the model’s architecture in terms of layers, nodes etc.?
- What are the limitations of UAV? Relatively small area? Other issue?
- What are the UAV characteristics, spectral bands? Similarity to satellite data, geospatial resolution?
Author Response
R: Dear reviewer,
We appreciate the revision and comments directed at our manuscript. We addressed each of them bellow.
1) “How data was collected? What is the dataset size?” Data was collected with in-field measurements and following agronomical standard procedures. More information regarding data collection and data-set size were detailed in the method section of our manuscript. Please see section 2.2
2) “What is the model’s architecture in terms of layers, nodes, etc.?” We have provided additional information regarding the settings of our models. As we used the library default values, we descrived the most important ones in the method section (please see section 2.4). The implemented open-source software Weka 3.9.4 was used.
3) “What are the limitations of UAV? Relatively small area? Other issues?” We believe that the major limitation of a UAV is the low capacity to compensate and analyze larger areas such as orbital imagery. However, it is important when considering the spatial resolution and highly detailed vision of the plantation itself; providing interesting information at the plant level. We described a new paragraph in the discussion section debating the difficulties, limitations, and implications of UAV-based imagery usage in our study as well as in others.
4) “What are the UAV characteristics, spectral bands? Similarity to satellite data, geospatial resolution?” We have further detailed the information regarding the spectral bands of the used UAV-embedded sensors in our method section (please see section 2.3).
We believe that your contribution was vital to help us improve our manuscript. Again, thank you very much for providing these questions.
Kind regards,
Authors.

Reviewer 2 Report
This manuscript is written in grace.
I cannot find anything wrong.
Rather, it contains exciting results with solid methods.
One thing I would suggest is that citing a paper which has a similar topic like following;
“Parmley K, Nagasubramanian K, Sarkar S, Ganapathysubramanian B, Singh AK. Development of optimized phenomic predictors for efficient plant breeding decisions using phenomic-assisted selection in soybean. Plant Phenomics. 2019 Jul 28;2019:5809404.”
Again, this manuscript is truly useful and interesting for readers including myself.
Author Response
R: Dear reviewer,
We are happy to hear of your positive feedback. We are flattered and appreciate that you took the time to read our manuscript. We have inserted the suggested reference. Thank you very much.
Kind regards,
Authors.
Reviewer 3 Report
Title “ with” “using???”
Line 15 are this results from the overall performance of the model? Missing “, respectively.” Units are missing.
1.9 % of error in LN is high. Under what criteria the authors consider this is a good performance.
Line 16 - what is the framework proposed? Seems like testing different ML methods is not a framework and authors should review that to avoid confusion.
Line 38, Nitrogen topic should be move along with the global production and how it depends on N.
Line 42. “be a problem” authors should be more specific
“plantation” replace by “crop”
Rare cases in which producer would apply N to a point of damaging the crop. Please review
Line 44 the problem is not very well explain and needs to be review. N can be limiting or not at the right time and rate. The authors should be more specific.
Line 45 where is this method use? How many producers use it?
Beyond the fertility decision what other uses could you get from leaf N concentration…? Crop modeling tools?
The author should explain how the leaf N tissue is utilized.
Line 50 how do you connect remote sensing with the N concentration. “the adoption of UAV-based images analysis may be of assistance to perform plant nutrient content and growth-status estimative.” Why?
Line 53-55 where is this common? What environments? Or are the authors talking specifically about new hybrids. I think the author are confusing a corn breeding program with field production and how to determine the N rate for a field.
Line 44 again I think the “ problem” is confusing … the rest of the introduction is focused on breeding and not N management.
Line 62-63 – not much content in that sentence. Use for what? what goal? “Predicting nutrient content and plant height with remote systems and intelligent methods is a goal for many researchers. This method consists of a low-cost and quick tool that has been applied in the recent decade for different crops.” = very vague . Same line 79-80
Line 86-87 the author are missing relevant literature on this topic – If the focus of the manuscript is crop nutrition and fertility this should be discussed and explain clearly.
For nutrient analysis, a recent study [42] implemented virtual reference methods
to perform N fertilization recommendation. ==> and ?
Line 79-90 this whole section doesn’t tell much to the reader about the relevance of this work and the key contribution. To lead into the objectives the gap of knowledge and the scope of the work should be correctly layout. The author mixed breeding, N management, ML, and more …without discussing why their work is important or unique.
Line 93 how has been this analyze in other manuscripts? And why the machine learning is central?
Line 99-102 not needed.
Whats the current perspective or advances of remote sensing into the breeding programs?
M&M
LNC method, give a brief summary of the technique
The measurements for leaf N and height were only taken once. Why?
“At 50 days after emergence (DAE), when the corn varieties were in full flowering, the average
123 third of five leaves for corn crop was collected in each experimental unit. The leaf nitrogen content
124 (LNC, g kg-1) was obtained by the methodology described in [44]. On this same date, plant height
125 (PH, m) was obtained by the average of five plants chosen at random in each plot. For this, a
126 measuring tape was used, positioned from the base of the plant to its apex.”
The fact that the authors are focusing on the flowering stage confirms that the manuscript is oriented to plant breeding
How did the authors process the images? Calibration? Details about the flight?
The information that the authors provide in the M&M is not sufficient to reproduce the study.
“Experimental protocol” this section should be data analysis
Line 174-175 no need to tell how the section will be organize – please remove from the manuscript.
Pearson correlation was not explain in the m&m –
How did the authors relate the N concentration and the remote sensing images?
How did the authors utilize the correlation matrix to limit the number of variables use?
Were the results consistent both years? What was the variability in N concentration among years?
Results
Figure 2 very small… hard to understand.
What is the machine learning architecture?
Adding MAE would help in the interpretation of the results.
Titles of each section could be improved. For example, model performance for …height? Leaf N?
Line 236-237 discussion text
Did the model perform better for different hybrids?
Table 4 – not sure where are the ML algorithms. What is the order? If same as rank from the text, please replace. Units are missing. What is the merit?
What are the units of RMSE in figure 4?
Figures need better axis titles LN RMSE (mg/kg)….
Line 218-219 higher…lower…. Authors should provide numbers and results rather that adjectives.
Line 217 what is larger K values?
Tables and figures are showing pretty much the same information. Authors could display an observed vs predicted figure with the statistics in a text box.
Remove tables
Results in general need work.
Line 194 what is topdressing condition?
Where is the overall performance of the model?
Are the figures from the 10-fold validation? Validation section should be expand and clarify
Discussion
Line 241-250 reads as a summary not like a discussion
Line 294 what is plant status? Plant height was used.
Line 297 – 299 the paper is focused on 2 variables. The authors did not make the connection with management.
Are the errors found in the predictions high, low, or medium and how do they affect the use of this tool?
Conclusions
Line 278 : machine learning evaluation ???? I think what the authors try to do was to predict the LN and PH by using different ML methods…. Please clarify
The statements such as “perform better” “it is a better method” etc. need to be supported with numbers and evidences from the results section. What is better?
“Regardless, the integration between both data improved their accuracy.” What data? Be more specific
Author Response
Dear reviewer,
Thank you very much for such an extensive and careful evaluation of our manuscript. We fully agreed with the concerns raised in your commentary and implement them in the new version of the manuscript. We also provided, in this letter, a response to each suggestion and concern expressed. We believe that these additions helped to improve the overall quality of our manuscript. We hope that you find them appropriate.
Title “ with” “using???”
R: Thank you for this suggestion. We correct it accordantly.
Line 15 are this results from the overall performance of the model? Missing “, respectively.” Units are missing.
R: Thank you for this suggestion. We correct it accordantly.
1.9 % of error in LN is high. Under what criteria the authors consider this is a good performance.
R: We are sorry for not clarifying it. The RMSE was 1.9 g.kg¹. Since our experiment had a high data variation, we considered this estimative appropriate. Thank you for this question.
Line 16 - what is the framework proposed? Seems like testing different ML methods is not a framework and authors should review that to avoid confusion.
R: Thank you for this suggestion. We correct it accordantly.
Line 38, Nitrogen topic should be move along with the global production and how it depends on N.
R: Thank you for this suggestion. We correct it accordantly.
Line 42. “be a problem” authors should be more specific
“plantation” replace by “crop”
Rare cases in which producer would apply N to a point of damaging the crop. Please review
R: Thank you for this suggestion. We correct it accordantly.
Line 44 the problem is not very well explain and needs to be review. N can be limiting or not at the right time and rate. The authors should be more specific.
R: Thank you for this suggestion. We correct it accordantly.
Line 45 where is this method use? How many producers use it?
Beyond the fertility decision what other uses could you get from leaf N concentration…? Crop modeling tools?
The author should explain how the leaf N tissue is utilized.
R: Thank you for this suggestion. We rewrite this sentence to help ascertain the relationship between N prediction and crop management.
Line 50 how do you connect remote sensing with the N concentration. “the adoption of UAV-based images analysis may be of assistance to perform plant nutrient content and growth-status estimative.” Why?
R: Thank you for this question. To better ascertain this relationship, we have inserted multiple references alongside our manuscript that are indicators of this observation.
Line 53-55 where is this common? What environments? Or are the authors talking specifically about new hybrids. I think the author are confusing a corn breeding program with field production and how to determine the N rate for a field.
R: Thank you for this observation. We have removed this paragraph since it has caused confusion regarding the theme of our manuscript.
Line 44 again I think the “ problem” is confusing … the rest of the introduction is focused on breeding and not N management.
R: R: Thank you for this observation. As we removed the previous paragraph, we believe now that the introduction is more appropriate to the evaluated issue in our experiment.
Line 62-63 – not much content in that sentence. Use for what? what goal? “Predicting nutrient content and plant height with remote systems and intelligent methods is a goal for many researchers. This method consists of a low-cost and quick tool that has been applied in the recent decade for different crops.” = very vague . Same line 79-80
R: Thank you for this suggestion. We correct it accordantly.
Line 86-87 the author are missing relevant literature on this topic – If the focus of the manuscript is crop nutrition and fertility this should be discussed and explain clearly.
For nutrient analysis, a recent study [42] implemented virtual reference methods
to perform N fertilization recommendation. ==> and ?
R: Thank you for pointing this out. We corrected the phrase and now believe that this paragraph is more appropriate to expose the aim of our research.
Line 79-90 this whole section doesn’t tell much to the reader about the relevance of this work and the key contribution. To lead into the objectives the gap of knowledge and the scope of the work should be correctly layout. The author mixed breeding, N management, ML, and more …without discussing why their work is important or unique. Line 93 how has been this analyze in other manuscripts? And why the machine learning is central?
R: Thank you for mentioning this. We performed the necessary modifications to better explain the novelty of our research. We also removed and corrected the layout of our introduction to be more relevant to our analysis. We believe that this helped us to better present our manuscript.
Line 99-102 not needed.
R: Thank you for this suggestion. We correct it accordantly.
Whats the current perspective or advances of remote sensing into the breeding programs?
R: Thank you for this question. Since we removed the only paragraph in our entire manuscript related to breeding, we believe that no more confusion related to this issue will happen. We did not focus on breeding programs and only used multiple cultivars to represent better the in-field variation encountered by farmers in the country. We believe now that the suggestions and questions raised helped us to improve our introduction section. Thank you very much.
M&M
LNC method, give a brief summary of the technique
R: Thank you for this suggestion. We correct it accordantly.
The measurements for leaf N and height were only taken once. Why?
R: Thank you for this question. They were taken once at the 50 DAE of each crop-season. Mainly because this frame is ideal to estimate both variables in this stage. We have added more information to the method section.
“At 50 days after emergence (DAE), when the corn varieties were in full flowering, the average
123 third of five leaves for corn crop was collected in each experimental unit. The leaf nitrogen content
124 (LNC, g kg-1) was obtained by the methodology described in [44]. On this same date, plant height
125 (PH, m) was obtained by the average of five plants chosen at random in each plot. For this, a
126 measuring tape was used, positioned from the base of the plant to its apex.”
The fact that the authors are focusing on the flowering stage confirms that the manuscript is oriented to plant breeding
R: Thank you for this commentary. We focused the analysis on the flowering stage because it is an essential epoch to estimate yield. Regardless, we understand that the text could have caused some confusion in the orientation of our research. We believe now that the corrections performed has helped us to better demonstrate the main objective of our study.
How did the authors process the images? Calibration? Details about the flight?
The information that the authors provide in the M&M is not sufficient to reproduce the study.
R: Thank you for observing this. We inserted this information into your methods section.
“Experimental protocol” this section should be data analysis
R: Thank you for this suggestion. We correct it accordantly.
Line 174-175 no need to tell how the section will be organize – please remove from the manuscript.
R: Thank you for this suggestion. We correct it accordantly.
Pearson correlation was not explain in the m&m –
R: Thank you for observing this. We have added this information in the method section.
How did the authors relate the N concentration and the remote sensing images?
R: Thank you for this question. We extracted the mean pixel values for each grid/plantation plot evaluated in our experimental area. With the respective mean pixel values of each plot, we used them as input data to estimate the measured in-field values of LNC and PH in their corresponding plot. We correct our manuscript with this information.
How did the authors utilize the correlation matrix to limit the number of variables use?
R: Thank you for this question. We did not implement a correlation matrix to limit the number of variables, but we did evaluate the different combinations based upon manually adjusting the number of variables used. We have added information on the method section that may help understand this issue. Thank you kindly.
Were the results consistent both years? What was the variability in N concentration among years?
R: Thank you for this question. Indeed, the results were consistent enough for both years and LNC and PH maintained proximal variances and coefficient of variance values. When calculating the variance for LNC, we obtained 21.33 g.kg-1 and 20.21 g.kg-1 in 2017/2018 and 2018/2019 crop seasons, respectably. As for PH, the variance was 0.107 m and 0.101 m for 2017/2018 and 2018/2019 crop seasons, respectably. We inserted this information in our manuscript. Thank you again.
Results
Figure 2 very small… hard to understand.
R: Thank you for pointing this out. We modified the layout of the figure as well as the manuscript.
What is the machine learning architecture?
R: Thank you for this question. We have added information in the method section describing the architecture of our models.
Adding MAE would help in the interpretation of the results.
R: Thank you for this suggestion. We have added MAE into a new figure.
Titles of each section could be improved. For example, model performance for …height? Leaf N?
R: Thank you for this suggestion. We correct it accordantly.
Line 236-237 discussion text
R: Thank you for this suggestion. We correct it accordantly.
Did the model perform better for different hybrids?
R: Thank you for noticing it. During our experimental phase, we did evaluate the model’s performances with different hybrid combinations. However, it did not return interesting results, and we opt to remove from the final version of our manuscript. Regardless we have added this information in the method section summarizing these initial experiments. Thank you kindly for asking.
Table 4 – not sure where are the ML algorithms. What is the order? If same as rank from the text, please replace. Units are missing. What is the merit?
R: Thank you for pointing this out. We performed some alterations in the text that helps to better explain this table. The merit score value was calculated as described in section 2.4. Thank you kindly.
What are the units of RMSE in figure 4?
R: Thank you for noticing it. They are in g.kg-1.
Figures need better axis titles LN RMSE (mg/kg)….
R: Thank you for this suggestion. We correct it accordantly.
Line 218-219 higher…lower…. Authors should provide numbers and results rather that adjectives.
R: Thank you for mentioning this. We have reevaluated our entire manuscript searching for these inconsistencies and performed the necessary corrections.
Line 217 what is larger K values?
R: Thank you for this observation. We meant “higher” values. We correct it accordantly.
Tables and figures are showing pretty much the same information. Authors could display an observed vs predicted figure with the statistics in a text box.
Remove tables
R: Thank you for this suggestion. We correct it accordantly.
Results in general need work.
Line 194 what is topdressing condition?
R: Thank you for pointing this out. The topdressing condition corresponds with the highest fertilization rate, 180 kg ha-1. We correct it accordantly.
Where is the overall performance of the model?
R: Thank you for this question. The overall performance of the best model (RF) presented an r = 0.91 and 0.86 and RMSE equal to 1.9 g.k-1 and 0.17 m, for both LNC and PH, respectively. We have added this information in both abstract, results, and conclusion sections.
Are the figures from the 10-fold validation? Validation section should be expand and clarify
R: The Figures are indeed from the 10-fold validation. In this regard, we have updated the information in our method section. Thank you kindly for pointing this out.
Discussion
Line 241-250 reads as a summary not like a discussion
R: Thank you for this observation. We remove this part from our manuscript.
Line 294 what is plant status? Plant height was used.
R: Thank you for this observation. We correct it accordantly.
Line 297 – 299 the paper is focused on 2 variables. The authors did not make the connection with management.
R: Thank you for this observation. We correct it accordantly.
Are the errors found in the predictions high, low, or medium and how do they affect the use of this tool?
R: Thank you for this question. The errors observed for the RF model can be considered relatively lower or similar to similar research. In this regard, we have added information into our discussion section. Thank you very much.
Conclusions
Line 278 : machine learning evaluation ???? I think what the authors try to do was to predict the LN and PH by using different ML methods…. Please clarify
R: Thank you for this suggestion. We correct it accordantly.
The statements such as “perform better” “it is a better method” etc. need to be supported with numbers and evidences from the results section. What is better?
R: Thank you for this observation. We corrected it by adding number results from our experiment.
“Regardless, the integration between both data improved their accuracy.” What data? Be more specific
R: Thank you for pointing this out. We have removed this phrase from the conclusion section.
We also performed an extensive and careful revision of the English language in our text, as well as asking for a native-speaker to evaluate it. We corrected grammar, structure sentences, and other errors that were decreasing the presentation of our manuscript.
Thank you very much. I hope that you consider the modifications suitable.
Kind regards,
Authors.

Reviewer 4 Report
Review manuscript remotesensing-864160, "Leaf Nitrogen Content and Plant Height Prediction for Corn-Plants with UAV-Based Multispectral Imagery and Machine Learning Techniques”.
This manuscript combines UAV imagery, derived vegetation indices with machine learning techniques to estimate leaf nitrogen content and plant height for corn. Although none of these elements is really new, the combination of these techniques makes the manuscript very interesting.
Although the English grammar should be improved at several places (please let a native English speaking editor have a look at the manuscript), the set-up of the manuscript is well-done and well-structured.
A major comment is that the authors frequently use the Pearson’s correlation coefficient (r), which is a measure for a linear relationship, whereas the used machine learning techniques are mostly non-linear techniques. So, it does not seem logical to use r as a measure for evaluating the performance of the different techniques. My guess is that r is calculated from a linear regression between all predicted and observed values. In this case r still can be very large even if there is a huge bias in the predicted values. It is much better to calculate the deviation from the 1:1 line. To illustrate this I would like to see some scatterplots of predicted against observed N content and of predicted against observed plant height. I hope the authors calculated the RMSE with respect to the 1:1 line and not with respect to the new regression line. The latter would be misleading.
So, the authors should present the RMSE with respect to the 1:1 line (illustrated also with a few scatter plots) and not the Pearson’s correlation coefficient.
Some further comments:
- The first sentence of the abstract should be reformulated, because it does not bring the intended content to the reader. Now stating “making it possible to estimate vegetative growth” suggests that this is only possible under ideal conditions of N supply. My guess is that this is not the intention of the authors. This also comes back in lines 51-52.
- In line 8-9 it is stated that the Sequoia was used in the first season and the MicaSense in the second one. However, according to section 2.3 it was the other way around. This inconsistency should be clarified.
- As a key word “shallow learner” is listed, although this term is not used in the manuscript at all. So, do not use this as a key word.
- Line 25: should this not be 101 million tons of production?
- Line 91: “unknown research” should be replaced by “no known research”.
- Section 3 should provide more details on the flights performed, like flight height, flight time, solar angle, cloud conditions, etcetera.
- Line 151-152: authors should use the past tense here instead of the present perfect tense.
- Line 161: in table 1 “Sunil et al.” should be “Saha et al.”
- Figure 2: The 2 levels of topdressing are represented in 2 different colours. However, many scatterplots clearly show 2 clusters of data points that are not the 2 levels of topdressing (blue and red dots). Where are the 2 clusters of data points coming from? Are these the 2 different years? If so, are there intercalibration issues between the different sensors used in both years? Authors should pay attention to this important issue of figure 2. A positive correlation between red reflectance and plant height and a negative correlation between NIR reflectance and plant height is different from most results in literature, so this should be explained further.
- The information provided in tables 2+3 and in figures 3+4 is basically the same, only the way of presentation is different. So, one of the two is redundant. Authors should either present the 2 tables or the 2 figures, not both.
- The authors should not mix up the symbols “r” and “R”.
Author Response
R: Dear reviewer,
Thank you very much for the detailed revision of our manuscript. We have answered point-by-point each statement of your review and implemented suggestions and recommendations described in the comments. We believe that your contribution has helped us to improve our manuscript quality.
1) We performed an extensive and careful revision of the English language in our text, as well as asking for a native-speaker to evaluate it. We have gone over the manuscript and made several corrections to spelling/grammar/structure errors. Thank you very much for this recommendation.
2) Indeed, r is calculated between all predicted and observed values. As suggested by the reviewer, we also calculated the RMSE of our predictions and inserted it in the form of a new figure in the results section. Again, thank you very much for this suggestion.
3) We have reformulated the introduction section of our abstract regarding the concerned issue. As well as change it in lines 51-52. Thank you very much.
4) We used Sequoia in the first crop-season and MicaSense RedEdge in the second crop-season. We have corrected this mistake by removing this phrase from the abstract since we needed to reduce the number of words. Regardless, thank you for observing this.
5) The term “shallow learner” was removed from the manuscript entirely. Thank you for this highly detailed observation.
6) Indeed, we have corrected this issue. Thank you, again.
7) We replaced the inaccurate sentence with “no known research”. Thank you very much for this suggestion.
8) We have detailed the flight conditions of our experiment in this section. Thank you for pointing this out.
9) We changed it to the past-sentence. Thank you.
10) We replaced the citation in Table 1 with the corrected reference.
11) We appreciate the suggestion for Figure 3 (old Figure 2) and add the following paragraph in the Discussion to justify the results found: " In some scatterplots in Figure 3, it is possible to notice two clusters formed for some variables. Corn is a C4 plant that increases its plant mass due to the rain that occurred during its cultivation. As we perform inter-calibration issues between the different sensors, these results can be attributed to climatic differences (soil-water-plant-atmosphere interaction) between one crop season and the other (see Figure 2). Besides, our reflectance values in the Red spectral band are directly linked (positive correlation) with the LNC, corroborating the structure (photosynthetic pigments and plant cell structure). The opposite occurred with the NIR due to the high reflectance being influenced by the water in the structure, with similar results in Green (although the latter due to the structure of carotenoids being formed together with aspects of chlorophyll). Thus, the negative correlation between VIs used in this work and the LNC can be explained by the presence of the NIR band in their equations. It is important to highlight the behavior of the variation of the variables in the scatterplot was similar. Thus, the variability obtained in the data is important to make the generated models more general and with greater capacity to predict LNC and PH.".
12) Thank you for this observation. Since Figures are a more stimulating manner for communicating our results, we have removed the Tables 1 and 2 from our manuscript.
13) We replaced the correlation coefficient (r) with the correct symbol in all text and figures of our manuscript.
Again, thank you very much for providing these comments. We believe that the contribution provided helped to improve this paper. We hope that you find the alterations appropriate.
Kind regards,
Authors.

Round 2
Reviewer 3 Report
Dear authors,
Below my suggestions and comments on your replies/changes. There were several comments that the authors said “we correct accordantly” without indicating how was this change, new lines, text added, etc.
Tittle : “Corn-Plants” replace by “Corn”
Keywords: needs review. The paper did not look at “nutrient”, and “precision agriculture” is not a key word for this manuscript.
Line 16 - what is the framework proposed? Seems like testing different ML methods is not a framework and authors should review that to avoid confusion.
R: Thank you for this suggestion. We correct it accordantly.
RR: In my opinion, a prediction that works at flowering is not suitable for N nutrient management. The authors should add more specific implications such as breading, etc. At flowering most of the N has been taken up by the plant. The author could consider irrigated systems where later applications are still feasible or in crop modeling applications.
Line 12 “r” “R” replace
Lines 48-50 : “The identification of N supply in plants is an important issue to adjust the correct fertilization rates of the crops and promote the adequate fertilization rate.”
N supply from? It is not clear. Is the author talking about the N status? These terms are not interchangeable.
Revise grammar in these Lines and the ones before. The new additions need revision.
Line 54: “intelligent methods is gaining attention in the agriculture remote sensing” -> please remove “remote sensing” at the end.
Line 84 “please remove “crop”
Line 85-86 please remove the new addition – “The combination of variables
and configurations implemented in our paper demonstrates a feasible approach to be adopted in this
type of analysis.” Vague.
Line 91 delete the new addition.
Line 74 delete “crop”, reader will know that corn is a crop.
Line 86-87 the author are missing relevant literature on this topic – If the focus of the manuscript is crop nutrition and fertility this should be discussed and explain clearly.
For nutrient analysis, a recent study [42] implemented virtual reference methods
to perform N fertilization recommendation. ==> and ?
R: Thank you for pointing this out. We corrected the phrase and now believe that this paragraph is more appropriate to expose the aim of our research.
RR: The author still don’t have a good understanding of the reference cited and it is not fully related with their topic of research. For example = “ For nutrient analysis, a study [36] implemented virtual reference methods to perform N fertilization recommendation.” Virtual reference is a practical approach for in-season nutrient management. Is a way to established the reference value for the sufficient index. This is usually calculated before V12 in corn. This text and reference is not related with ML methods…
In addition, the next = “A revision study on yield and nitrogen status in precision agriculture applications
concluded that recent advances in remote sensing technologies and machine learning techniques
will result in more cost-effective and comprehensive solutions for the better crop state assessment.” Is not related with the previous virtual reference method….This is a clear evidence that the introduction needs more revisions to target the clear and novel application of the method proposed for the authors.
Line 79-90 this whole section doesn’t tell much to the reader about the relevance of this work and the key contribution. To lead into the objectives the gap of knowledge and the scope of the work should be correctly layout. The author mixed breeding, N management, ML, and more …without discussing why their work is important or unique. Line 93 how has been this analyze in other manuscripts? And why the machine learning is central?
R: Thank you for mentioning this. We performed the necessary modifications to better explain the novelty of our research. We also removed and corrected the layout of our introduction to be more relevant to our analysis. We believe that this helped us to better present our manuscript.
RR: please cited all the changes done in response to my previous comments. It is not clear and hard to find the substantial changes addressing this gap in the MS.
What’s the current perspective or advances of remote sensing into the breeding programs?
R: Thank you for this question. Since we removed the only paragraph in our entire manuscript related to breeding, we believe that no more confusion related to this issue will happen. We did not focus on breeding programs and only used multiple cultivars to represent better the in-field variation encountered by farmers in the country. We believe now that the suggestions and questions raised helped us to improve our introduction section. Thank you very much.
RR: what is then the focus of the paper and applicability? If the authors are focusing on nutrient management, again, prediction at flowering is not as useful.
Second, the variation in hybrids is probably to created variability in the N tissue concentration rather than trying to be representative of the entire country.
I would argue with the authors that the nitrogen concentration and height are important for phenotyping and breeding programs.
In general, the introduction talks about N concentration. The authors should discuss why prediction of height is important and its application. How can be a prediction at flowering for height be used? …
It has to be clear why the machine learning technique bring something novel into the N concentration prediction – remote sensing for leaf N has been used in the last 30 years or more…
Line 88 “evaluate” -> to predict! The authors did’t evaluate the PH and leaf N …
Based on a comment in the m&m section the authors claimed to measure leaf N and height to estimate yield. This is not mention at all in the introduction. This point this needs to be clarify.
M&M
The measurements for leaf N and height were only taken once. Why?
R: Thank you for this question. They were taken once at the 50 DAE of each crop-season. Mainly because this frame is ideal to estimate both variables in this stage. We have added more information to the method section.
RR: the authors should justify why at flowering. Why this stage was preferred for the authors when they claimed to be
Line 118-124 , the new additions need grammar editing. For example at the is “respectively”…
“in-field of both LNC and PH.” -> please check English.
Line 120 “consistent enough” please clarify…
“proximal “ replace by “similar’ ?????
What’s the difference between “proximal variances” and “coefficient of variance values” …???
“At 50 days after emergence (DAE), when the corn varieties were in full flowering, the average
123 third of five leaves for corn crop was collected in each experimental unit. The leaf nitrogen content
124 (LNC, g kg-1) was obtained by the methodology described in [44]. On this same date, plant height
125 (PH, m) was obtained by the average of five plants chosen at random in each plot. For this, a
126 measuring tape was used, positioned from the base of the plant to its apex.”
The fact that the authors are focusing on the flowering stage confirms that the manuscript is oriented to plant breeding
R: Thank you for this commentary. We focused the analysis on the flowering stage because it is an essential epoch to estimate yield. Regardless, we understand that the text could have caused some confusion in the orientation of our research. We believe now that the corrections performed has helped us to better demonstrate the main objective of our study.
RR: If the focus of the paper is to estimate YIELD! Then the introduction has to be reformulated.
There are multiple methods for predicting yield and those have not been introduced or discussed. This is a critical point to understand the logic of the manuscript.
Line 196 delete ‘later”.
Figure 3 caption need grammar revision. The English needs revision.
Pearson correlation was not explained in the m&m –
R: Thank you for observing this. We have added this information in the method section.
RR: Line 161-162 “To ensure how well the predictions went” this needs further explanation. The Pearson correlation is not to evaluate the predictions. But to explore relationship between variables…. The RMSE, R2, MAE those are statistical ways to evaluate the model performance.
Line 165 delete “also”. The obs vs pred plot is to evaluate the model performance…please clarify. Review English grammar.
How did the authors utilize the correlation matrix to limit the number of variables use?
R: Thank you for this question. We did not implement a correlation matrix to limit the number of variables, but we did evaluate the different combinations based upon manually adjusting the number of variables used. We have added information on the method section that may help understand this issue. Thank you kindly.
RR: why the authors adjusted the number of variables manually? and they did not let the ML make the model selection.
Results
Figure 2 very small… hard to understand.
R: Thank you for pointing this out. We modified the layout of the figure as well as the manuscript.
RR still pretty small and hard to read. Review caption.
What is the machine learning architecture?
R: Thank you for this question. We have added information in the method section describing the architecture of our models.
RR: where? Please bring into the reply. Thanks
Figure 6 caption is not complete. Please review. Axis font labels need review. Missing units.
Line 263 “Figure 6” or figure 7?
Please change actual in the axis labels for “observed”
Figure 7 is missing the statistical parameters , units, please improve the figure. Review caption. For example, relationship between predicted and observed xxxx lean n concentration (LNC) and plant height (PH). Authors should check the quality of all figures and captions.
Missing model fitting?
Line 229: “estimative” ? -> replace by prediction.
What are the units of RMSE in figure 4?
R: Thank you for noticing it. They are in g.kg-1.
RR: Please fix the font on the units, “r” -> “pearson’s correlation” . For the caption RMSE has to be explained root mean square….(RMSE)… ETC.
Line 194 what is topdressing condition?
R: Thank you for pointing this out. The topdressing condition corresponds with the highest fertilization rate, 180 kg ha-1. We correct it accordantly.
RR: both treatments were top dressing….
Line 210 – please clarify how they were different the correlations at different top dressing treatments?
“each N content topdressing condition.” => it is not a condition it is a treatment the amount of N , and leaf N is a N status. Please be consistent with the terms.
Discussion
Line 283-284 “and …”and” please correct.
Line 284-285 …the two clusters are the treatment effect. Not clear why the authors are pointing this out in this section of the discussion. And? What are the authors trying to say with this?
“As we perform inter-calibration issues between the different sensors, these results can be attributed to climatic differences” ????? not sure what this means and how this is related with the discussion of a particular result.
Line 297 “predict phenotype”. Please demonstrate how are you predicting phenotype…
Line 285-286 – is totally out of context…please revise
Line 294 what is plant status? Plant height was used.
R: Thank you for this observation. We correct it accordantly.
RR: what line? Please bring the replies here.
Line 297 – 299 the paper is focused on 2 variables. The authors did not make the connection with management.
R: Thank you for this observation. We correct it accordantly.
RR: where? Please bring the lines, or section where you address this issue. Thanks
Are the errors found in the predictions high, low, or medium and how do they affect the use of this tool?
R: Thank you for this question. The errors observed for the RF model can be considered relatively lower or similar to similar research. In this regard, we have added information into our discussion section. Thank you very much.
RR: please bring the lines and part of the MS changed to address this issue. Thanks
Lines:
344 Indeed, the combination of multiple variables and configurations implemented in our paper
345 demonstrates a feasible approach to be adopted in this type of analysis. More than often, previous
346 research focused on the evaluation of a single variable our cultivar, in a single period.
RR: These text doesn’t make sense, plus needs English revision. “Indeed” doesn’t fit there…. “More than often”….several places need to be reviewed. “This type of analysis” ….vague argument.
Lines”:
352 these conditions are essential to corroborate the robustness of the proposed approach with the
353 machine learning approach. Approach is repeated twice.
Conclusions
We also performed an extensive and careful revision of the English language in our text, as well as asking for a native-speaker to evaluate it. We corrected grammar, structure sentences, and other errors that were decreasing the presentation of our manuscript.
RR: please review your new text additions, figures, captions.
Author Response
Dear authors,
Below my suggestions and comments on your replies/changes. There were several comments that the authors said “we correct accordantly” without indicating how was this change, new lines, text added, etc.
Dear reviewer,
We are sorry for not indicating how all the changes were conducted in the previous revision, as we only focused on major ones to not extend the length of our letter. We hope that this innocent mistake can be overlooked now that we, in this second letter, are providing a more complete response to your careful revision. Bellow are our point-by-point answers. Thank you, again, for evaluating our manuscript.
Title: “Corn-Plants” replace by “Corn”
R: We replaced “Corn-Plants” by “Corn” in our title.
Keywords: needs review. The paper did not look at “nutrient”, and “precision agriculture” is not a key word for this manuscript.
R: We exchanged the keywords for ones more related to our research. Thank you.
Line 16 - what is the framework proposed? Seems like testing different ML methods is not a framework and authors should review that to avoid confusion.
R: Thank you for this suggestion. We correct it accordantly.
RR: In my opinion, a prediction that works at flowering is not suitable for N nutrient management. The authors should add more specific implications such as breading, etc. At flowering most of the N has been taken up by the plant. The author could consider irrigated systems where later applications are still feasible or in crop modeling applications.
RRR: Originally, the intention behind the usage of the word “framework” in the previous version was to demonstrate to the reader a viable approach to process UAV-based data with different spectral bands and indices configurations sets and machine learning models to predict LNC and PH in plants such as corn. Regardless, we have removed the line 16 from the original manuscript to avoid the confusion. As for the prediction at the flowering stage,
Line 12 “r” “R” replace
R: We are currently using “r” as a correlation coefficient all over the manuscript since “R” is being used as an abbreviation for the Red Band in the method section.
Lines 48-50: “The identification of N supply in plants is an important issue to adjust the correct fertilization rates of the crops and promote the adequate fertilization rate.”
N supply from? It is not clear. Is the author talking about the N status? These terms are not interchangeable.
Revise grammar in these Lines and the ones before. The new additions need revision.
R: We changed “N supply” to “N content”. Thank you for noticing this. We also reviewed the line before this one (abstract section, line 1) where “N supply” expression was also used. We corrected it accordantly.
Line 54: “intelligent methods is gaining attention in the agriculture remote sensing” -> please remove “remote sensing” at the end.
R: “remote sensing” removed.
Line 84 “please remove “crop”
R: “crop” word removed.
Line 85-86 please remove the new addition – “The combination of variables and configurations implemented in our paper demonstrates a feasible approach to be adopted in this type of analysis.” Vague.
R: The addition was removed.
Line 91 delete the new addition.
R: The addition in this line was deleted.
Line 74 delete “crop”, reader will know that corn is a crop.
R: Understood. The word “crop” was removed from this line.
Line 86-87 the author are missing relevant literature on this topic – If the focus of the manuscript is crop nutrition and fertility this should be discussed and explain clearly.
R: We understand the concern raised regarding nutrition and fertility, mainly because of our writing choices in the past. But upon a careful revision and incorporation of your suggestions, we believe that we now have a reliable version of our introduction. We will be explaining them in more detail alongside this letter. To avoid repetition, we will summarize here that our approach was only to demonstrate a feasible alternative to predict LNC and PH with machine learning models and UAV-based spectral imagery. We also choose the flowering stage mainly because, from a methodological point-of-view, we aimed to evaluate the N content at a stage (flowering) were the plant demanded more N to produce their grains. We explain in detail the reason behind this in the subsequent commentary. However, we understand the raised issue and are sorry for the inconvenience caused. Because of that, we tried to remove as much as possible information on the introduction that could indicate that we aimed to improve fertilization recommendations. As for relevant literature, since we aimed for nitrogen, growth, and UAV-based imagery evaluated with machine learning models, we believe that we already inserted important references in the current version of our introduction. I’ll be listing them here shortly:
Wang, S.; Azzari, G.; Lobell, D.B. Crop type mapping without field-level labels: Random forest transfer and unsupervised clustering techniques. Remote Sens. Environ. 2019, 222, 303–317, doi:10.1016/j.rse.2018.12.026.
Hunt, E.R.; Daughtry, C.S.T. What good are unmanned aircraft systems for agricultural remote sensing and precision agriculture? Int. J. Remote Sens. 2018, 39, 5345–5376, doi:10.1080/01431161.2017.1410300.
Osco, L.P.; Marques Ramos, A.P.; Saito Moriya, É.A.; de Souza, M.; Marcato Junior, J.; Matsubara, E.T.; Imai, N.N.; Creste, J.E. Improvement of leaf nitrogen content inference in Valencia-orange trees applying spectral analysis algorithms in UAV mounted-sensor images. Int. J. Appl. Earth Obs. Geoinf. 2019, 83, 101907, doi:10.1016/j.jag.2019.101907.
Osco, L.P.; Ramos, A.P.M.; Pinheiro, M.M.F.; Moriya, É.A.S.; Imai, N.N.; Estrabis, N.; Ianczyk, F.; de Araújo, F.F.; Liesenberg, V.; de Castro Jorge, L.A.; et al. A machine learning approach to predict nutrient content in valencia-orange leaf hyperspectral measurements. Remote Sens. 2020, 12, doi:10.3390/rs12060906.
Song, Y.; Wang, J. Soybean canopy nitrogen monitoring and prediction using ground based multispectral remote sensors. Int. Geosci. Remote Sens. Symp. 2016, 2016-Novem, 6389–6392, doi:10.1109/IGARSS.2016.7730670.
Cammarano, D.; Fitzgerald, G.J.; Casa, R.; Basso, B. Assessing the robustness of vegetation indices to estimate wheat N in mediterranean environments. Remote Sens. 2014, 6, 2827–2844, doi:10.3390/rs6042827.
Osco, L.P.; Ramos, A.P.M.; Pereira, D.R.; Moriya, érika A.S.; Imai, N.N.; Matsubara, E.T.; Estrabis, N.; de Souza, M.; Junior, J.M.; Gonçalves, W.N.; et al. Predicting canopy nitrogen content in citrus-trees using random forest algorithm associated to spectral vegetation indices from UAV-imagery. Remote Sens. 2019, 11, 1–17, doi:10.3390/rs11242925.
Zheng, H.; Li, W.; Jiang, J.; Liu, Y.; Cheng, T.; Tian, Y.; Zhu, Y.; Cao, W.; Zhang, Y.; Yao, X. A comparative assessment of different modeling algorithms for estimating leaf nitrogen content in winter wheat using multispectral images from an unmanned aerial vehicle. Remote Sens. 2018, 10, doi:10.3390/rs10122026.
Liu, Y.L.; Lyu, Q.; He, S.L.; Yi, S.L.; Liu, X.F.; Xie, R.J.; Zheng, Y.; Deng, L. Prediction of nitrogen and phosphorus contents incitrus leavesbased onhyperspectral imaging. Int. J. Agric. Biol. Eng. 2015, 8, 80–88, doi:10.3965/j.ijabe.20150802.1464.
Maxwell, A.E.; Warner, T.A.; Fang, F. Implementation of machine-learning classification in remote sensing: An applied review. Int. J. Remote Sens. 2018, 39, 2784–2817, doi:10.1080/01431161.2018.1433343.
Osco, L.P.; Ramos, A.P.M.; Moriya, É.A.S.; Bavaresco, L.G.; de Lima, B.C.; Estrabis, N.; Pereira, D.R.; Creste, J.E.; Júnior, J.M.; Gonçalves, W.N.; et al. Modeling hyperspectral response of water-stress induced lettuce plants using artificial neural networks. Remote Sens. 2019, 11, doi:10.3390/rs11232797.
Chlingaryan, A.; Sukkarieh, S.; Whelan, B. Machine learning approaches for crop yield prediction and nitrogen status estimation in precision agriculture: A review. Comput. Electron. Agric. 2018, 151, 61–69, doi:10.1016/j.compag.2018.05.012.
Han, L.; Yang, G.; Dai, H.; Xu, B.; Yang, H.; Feng, H.; Li, Z.; Yang, X. Modeling maize above-ground biomass based on machine learning approaches using UAV remote-sensing data. Plant Methods 2019, 15, 1–19, doi:10.1186/s13007-019-0394-z.
Yang, N.; Liu, D.; Feng, Q.; Xiong, Q.; Zhang, L.; Ren, T.; Zhao, Y.; Zhu, D.; Huang, J. Large-scale crop mapping based on machine learning and parallel computation with grids. Remote Sens. 2019, 11, 1–22, doi:10.3390/rs11121500.
For nutrient analysis, a recent study [42] implemented virtual reference methods to perform N fertilization recommendation. ==> and ?
R: Thank you for pointing this out. We corrected the phrase and now believe that this paragraph is more appropriate to expose the aim of our research.
RR: The author still don’t have a good understanding of the reference cited and it is not fully related with their topic of research. For example = “ For nutrient analysis, a study [36] implemented virtual reference methods to perform N fertilization recommendation.” Virtual reference is a practical approach for in-season nutrient management. Is a way to established the reference value for the sufficient index. This is usually calculated before V12 in corn. This text and reference is not related with ML methods…
RRR: Upon reevaluating your statement, we fully agreed with and choose to remove this phrase and reference to avoid confusion. Thank you.
In addition, the next = “A revision study on yield and nitrogen status in precision agriculture applications concluded that recent advances in remote sensing technologies and machine learning techniques will result in more cost-effective and comprehensive solutions for the better crop state assessment.” Is not related with the previous virtual reference method…
This is a clear evidence that the introduction needs more revisions to target the clear and novel application of the method proposed for the authors.
R: We agreed with this observation and choose to reallocate this affirmation into the beginning of the paragraph.
Original: “The combination of machine learning techniques and VIs is an important subject in agricultural applications. As no consent over the most reliable method nor spectral index has been achieved for particular tasks, the aforementioned researchers proposed different approaches to be implemented in their analysis. However, the RF model has been highly evaluated in recent years to help solve multiple related issues. For corn, one study [34] estimated above‑ground biomass of cornfields with both UAV-based image and machine learning methods and determined that the RF model gave the most balanced results overall. Another research [35] that aimed to detect corn-fields and other crops in images also stated that the RF learner reached high accuracies in a time-series evaluation. For nutrient analysis, a study [36] implemented virtual reference methods to perform N fertilization recommendation. A revision study on yield and nitrogen status in precision agriculture applications [37] concluded that recent advances in remote sensing technologies and machine learning techniques will result in more cost-effective and comprehensive solutions for the better crop state assessment.”
Current Lines 69-79: “A revision study on yield and N content in precision agriculture applications [34] concluded that recent advances in remote sensing technologies and machine learning techniques will result in more cost-effective and comprehensive solutions for the better crop state assessment. The combination of machine learning techniques and VIs is an important subject in agricultural applications. As no consent over the most reliable method nor spectral index has been achieved for particular tasks, the aforementioned researchers proposed different approaches to be implemented in their analysis. However, the RF model has been highly evaluated in recent years to help solve related issues. For corn, one study [35] estimated above‑ground biomass of cornfields with both UAV-based image and machine learning methods and determined that the RF model gave the most balanced results overall. Another research [36] that aimed to detect corn-fields and other crops in images also stated that the RF learner reached high accuracies in a time-series evaluation.”
Line 79-90 this whole section doesn’t tell much to the reader about the relevance of this work and the key contribution. To lead into the objectives the gap of knowledge and the scope of the work should be correctly layout. The author mixed breeding, N management, ML, and more …without discussing why their work is important or unique. Line 93 how has been this analyze in other manuscripts? And why the machine learning is central?
R: Thank you for mentioning this. We performed the necessary modifications to better explain the novelty of our research. We also removed and corrected the layout of our introduction to be more relevant to our analysis. We believe that this helped us to better present our manuscript.
RR: please cited all the changes done in response to my previous comments. It is not clear and hard to find the substantial changes addressing this gap in the MS.
RRR: We are sorry that we did not bring in the letter the substantial changes regarding this issue. However, since we modified it in this novel version, allow us to bring here, in this letter, the essential part in which we believe brings the main scope of our study. As you can see, we added a paragraph in which we explain the importance of considering our approach to evaluate N in corn plants with ML.
Original: “To the best of our knowledge, there is no known research involving the evaluation of both LNC in corn-crops and plant height (PH) using only aerial imagery processed by artificial intelligent techniques. Here, we present a reproducible approach to predict LNC and PH in corn-crops with machine learning algorithms and UAV-based multispectral imagery. The combination of variables and configurations implemented in our paper demonstrates a feasible approach to be adopted in this type of analysis. In this paper the following questions are addressed: 1) which machine learning models are most suitable to evaluate LNC and PH in corn plants with spectral data from UAV-based image? and; 2) Amongst all predictor variables (spectral indices, bands, and the combination of both), which one is the most useful for mapping LNC and PH based on the machine learning approach? The rest of this paper details the outcomes of these questions.”
Current Lines 80-95: “Under ideal conditions of N, corn plants can grow to their full potential reaching maximum height [37,38]. Considering that in a corn breeding program, multiple genotypes are evaluated annually, implementing different approaches to estimate its height and N content with UAV-based remote systems is essential to optimize monitoring of these areas. Currently, one of the main objectives of corn breeding programs is to identify genotypes with high-efficiency in N usage [39,40]. Obtaining rapid predictions with an alternative approach like machine learning and UAV-based imagery may enable breeding programs to evaluate multiple genotypes each year, allowing them to optimize the selection of the most promising plants in relation to N use efficiency.
As machine learning approaches have been proved [28,29, 31-33] to be a robust approach to evaluate heterogeneous data, it could return important results when considering different genotypes of corn plants. To the best of our knowledge, there is no known research involving the evaluation of both LNC in corn-crops and plant height (PH) using only aerial imagery processed by artificial intelligent techniques. Here, we present a reproducible approach to predict LNC and PH in corn with machine learning algorithms and UAV-based multispectral imagery. In this paper the following questions are addressed: 1) which machine learning models are most suitable to predict LNC and PH in corn plants with spectral data from UAV-based image? and, 2) Amongst all predictor variables (spectral indices, bands, and the combination of both), which one is the most useful for mapping LNC and PH based on the machine learning approach?”
What’s the current perspective or advances of remote sensing into the breeding programs?
R: Thank you for this question. Since we removed the only paragraph in our entire manuscript related to breeding, we believe that no more confusion related to this issue will happen. We did not focus on breeding programs and only used multiple cultivars to represent better the in-field variation encountered by farmers in the country. We believe now that the suggestions and questions raised helped us to improve our introduction section. Thank you very much.
RR: what is then the focus of the paper and applicability? If the authors are focusing on nutrient management, again, prediction at flowering is not as useful.
RRR: Dear reviewer, as stated, we caused some confusion during the last revision regarding this issue. We are deeply sorry for this inconvenience and believe that the information and revision shared between us in the last days have helped us to improve the aim of our approach. As explained, the objective of our manuscript was mainly to demonstrate a method to predict LNC and PH with machine learning models and UAV-based imagery. The date in which we collected the images was mostly because, from a methodological point-of-view, we wanted this information to be evaluated during the flowering period to estimate N content in an important stage for breeding programs. We understand that this was overlooked in the last version of our manuscript and letter, and hope that now this issue has been properly fixed. Thank you very much for helping us. Please, as we continue, read the rest of our responses as we believe will help in your judgment.
Second, the variation in hybrids is probably to created variability in the N tissue concentration rather than trying to be representative of the entire country.
I would argue with the authors that the nitrogen concentration and height are important for phenotyping and breeding programs.
In general, the introduction talks about N concentration. The authors should discuss why prediction of height is important and its application. How can be a prediction at flowering for height be used? …
It has to be clear why the machine learning technique bring something novel into the N concentration prediction – remote sensing for leaf N has been used in the last 30 years or more…
R: We are sorry for the confusion. Concerning plant height prediction in corn, we, unfortunately, could not find reliable research that used machine learning models and UAV-based imagery. This is the main reason why we were not able to provide much literature review on this matter. As for plant height information on the flowering stage, indeed it is not a topic that we focused because, from what we understand, plant height is information more essential to monitoring practices than to breeding. One example in corn that could benefit from plant height knowledge if mechanical harvesting. Regarding the usage of machine learning for N, we believe that the lines bellow can explain these issues. The main importance of machine learning in this aspect is to deal with different types of data (i.e. different year, sensors, cultivars, etc.) in the same model, learning from a pattern and executing this task with, generally, higher accuracy than traditional statistical methods. This information is also sustained by literature already referenced in the introduction, which we listed here previously.
Lines 64-68: “Different machine learning algorithms like random forests (RF), decision trees (DT), artificial neural network (ANN), support vector machines (SVM), among many others, have been adopted to attend various applications in agriculture remote sensing [6,31–33]. As demonstrated by the aforementioned studies, machine learning has helped to increase not only prediction’s accuracy of agronomic variables but also by solving complex problems related to data heterogeneity.”
Lines 89-91: “As machine learning approaches have been proved [28,29, 31-33] to be a robust approach to evaluate heterogeneous data, it could return important results when considering different genotypes of corn plants and N conditions.”
Line 88 “evaluate” -> to predict! The authors did’t evaluate the PH and leaf N …
R: Indeed, we predicted PH and LNC, not evaluated it. Thank you for correcting it.
Based on a comment in the m&m section the authors claimed to measure leaf N and height to estimate yield. This is not mention at all in the introduction. This point this needs to be clarify.
R: This is not the intent of our manuscript. We modified this statement to clarify this mistake. Additional information is reported in this specific commentary bellow. Thank you for pointing this.
M&M
The measurements for leaf N and height were only taken once. Why?
R: Thank you for this question. They were taken once at the 50 DAE of each crop-season. Mainly because this frame is ideal to estimate both variables in this stage. We have added more information to the method section.
RR: the authors should justify why at flowering. Why this stage was preferred for the authors when they claimed to be
RRR: We appreciate the concern expressed and fully understand it. In this regard, the evaluation of N in this period (50 DAE) is a well-established methodology in the scientific literature (I’ll be providing some references in this letter to sustain this information), especially in the area of plant nutrition. This period is when the corn plants are in full development and demands more N to produce their grains. We understand the concern about the applications on nutritional management and do agree with the raised issue. However, as we understand, the importance of building models that allow estimating N levels in this period (50 DAE) using VIs could make it possible to select more efficient genotypes regarding the use of this nutrient.
References:
- Almeida, V. C., Viana, J. M. S., DeOliveira, H. M., Risso, L. A., Ribeiro, A. F. S., & DeLima, R. O. (2018). Genetic diversity and path analysis for nitrogen use efficiency of tropical popcorn (Zea mays ssp. everta) inbred lines in adult stage. Plant Breeding, 137(6), 839–847. https://doi.org/10.1111/pbr.12650
- Torres, L. G., Rodrigues, M. C., Lima, N. L., Horta Trindade, T. F., Fonseca e Silva, F., Azevedo, C. F., & DeLima, R. O. (2018). Multi-trait multi-environment Bayesian model reveals g x e interaction for nitrogen use efficiency components in tropical maize. PLoS ONE, 13(6), 1–15. https://doi.org/10.1371/journal.pone.0199492
Line 118-124, the new additions need grammar editing. For example at the is “respectively” …
R: They were changed to “respectively”. Thank you for noticing.
“in-field of both LNC and PH.” -> please check English.
R: We corrected it to “176 in-field observations of LNC and PH”.
Line 120 “consistent enough” please clarify…
“proximal “ replace by “similar’ ?????
What’s the difference between “proximal variances” and “coefficient of variance values” …???
R: We exchanged the following sentence: “The measured values were consistent enough for both years and LNC and PH maintained proximal variances and coefficient of variance values.” To this one: “The measure mean-values of LNC and PH, for both seasons (2017/2018 and 2018/2019, did not result in statistical differences at a p-value under 0.05. For this, we performed a Shapiro-Wilk test followed by a pairwise t student test.” We believe that this modification in the text is a better explanation as to why both seasons did not differentiate from each other concerning the two years of analysis.
“At 50 days after emergence (DAE), when the corn varieties were in full flowering, the average third of five leaves for corn crop was collected in each experimental unit. The leaf nitrogen content (LNC, g kg-1) was obtained by the methodology described in [44]. On this same date, plant height (PH, m) was obtained by the average of five plants chosen at random in each plot. For this, a measuring tape was used, positioned from the base of the plant to its apex.”
The fact that the authors are focusing on the flowering stage confirms that the manuscript is oriented to plant breeding
R: Thank you for this commentary. We focused the analysis on the flowering stage because it is an essential epoch to estimate yield. Regardless, we understand that the text could have caused some confusion in the orientation of our research. We believe now that the corrections performed has helped us to better demonstrate the main objective of our study.
RR: If the focus of the paper is to estimate YIELD! Then the introduction has to be reformulated.
There are multiple methods for predicting yield and those have not been introduced or discussed. This is a critical point to understand the logic of the manuscript.
RRR: Again, we are sorry for misinterpreting the previous letter. We removed every “yield” word-sentence that could indicate that our manuscript aimed at it. Thank you for pointing this.
Line 196 delete ‘later”.
R: The word “later” was deleted.
Figure 3 caption need grammar revision. The English needs revision.
R: We revised Figure 3 caption. The previous version was: “Pearson's correlation between the variables evaluated for the condition of high (blue color) and low (red color) N content. The lower triangle corresponds with the dispersion graphics of each correlation in a pairwise manner. The higher triangle displays the value of each correlation. The main diagonal illustrates a kernel density-based histogram between the two N rates and the distribution of it according to the respective indices. The top row indicated box-plots between the distribution of values for each index according to the associated N content and, lastly, the first column performs a comparison between the basic distributions of data across the two N conditions.”.
And now it is: “Figure 3. Pearson's correlation (r) between each variable implemented in this study. The lower triangle corresponds with the dispersion graphics of each correlation in a pairwise manner. The higher triangle displays the correlation value (both the total values and also when analyzing separately low (red) and high (blue) topdressing conditions). The main diagonal illustrates a kernel density-based histogram between the two N rates and the distribution of it according to the respective variables. The top row displays box-plots of the values according to the associated N content and, the first column performs a comparison between the basic distributions of data across the two N conditions.”.
Pearson correlation was not explained in the m&m –
R: Thank you for observing this. We have added this information in the method section.
RR: Line 161-162 “To ensure how well the predictions went” this needs further explanation. The Pearson correlation is not to evaluate the predictions. But to explore relationship between variables…. The RMSE, R2, MAE those are statistical ways to evaluate the model performance.
RRR: Thank you for this observation. We have modified the text. From: “To ensure how well the predictions went, we evaluated the Pearson’s correlation coefficient and Root Mean Square Error (RMSE) over 176 samples extracted from the field plots considering the two crop seasons (2017/2018 and 2018/2019).”
To: “To explore the relationship between the predicted and the observed variables, we used Pearson’s correlation coefficient. To evaluate the model performance during the training phase, we calculated their respective Root Mean Square Errors (RMSE) over 176 samples extracted from the field plots considering the two crop seasons (2017/2018 and 2018/2019).”
Line 165 delete “also”. The obs vs pred plot is to evaluate the model performance…please clarify. Review English grammar.
R: We removed the word “also”. Since this phrase was related to the removed Figure 7, we exchanged it to be related to the new residual plot. Previously the phrase was: “Lastly, we also evaluated the dispersion of the predicted values compared against the measure in-field data with scatter plots.”. Now it is: “Lastly, we evaluated the dispersion of the residual values of the overall best prediction models for LNC and PH.”.
How did the authors utilize the correlation matrix to limit the number of variables use?
R: Thank you for this question. We did not implement a correlation matrix to limit the number of variables, but we did evaluate the different combinations based upon manually adjusting the number of variables used. We have added information on the method section that may help understand this issue. Thank you kindly.
RR: why the authors adjusted the number of variables manually? and they did not let the ML make the model selection.
RRR: The variables, in this case, are the spectral bands and spectral indices. We say “manually” simply because we did not use automated functions to divide which configuration set each of these variables belongs to. So, we, ourselves, allocated the spectral bands to one configuration set, allocated the spectral indices to the other configuration, and unified both sets into a third configuration set.
Results
Figure 2 very small… hard to understand.
R: Thank you for pointing this out. We modified the layout of the figure as well as the manuscript.
RR still pretty small and hard to read. Review caption.
RRR: We improved Figure 2 (now Figure 3) layout font and rewrite its caption. Thank you.
What is the machine learning architecture?
R: Thank you for this question. We have added information in the method section describing the architecture of our models.
RR: where? Please bring into the reply. Thanks
RRR: In the new version of the manuscript, the added paragraph describing the architectures of our models is located between lines 181 and 189, as follows:
“All models were initialized using the Weka 3.9.4 default library parameters and run over an Intel® Core™ i7 CPU and 12Gb RAM workstation. We considered the library default values for the number and depth of trees, nodes, and leaves in our decision tree models, as well as a different number of neighbors (1, 5, and 10) for the KNN algorithm. As stated, we considered two functions (RBF and polynomial) for the SVM learner, represented by exp(-gamma*|u-v|²) and (gamma*u'*v + coef0) ², respectively. Each value regarding the described variables was set to be calculated automatically considering the overall best predictions with an epsilon loss curve equal to 0.1. Last, we used a grid search approach to fine-tuning our linear regression model (RBF Regression), thus performing an hyperparametrization of this particular model-based.”
Complementary information was described in the previous paragraph, where we stated the functions and primary parameters of our models. Thank you for asking this.
Figure 6 caption is not complete. Please review. Axis font labels need review. Missing units.
R: We have corrected the font labels by adding the missing units.
Line 263 “Figure 6” or figure 7?
R: In this new version, we modified this line and now it is Figure 7. Thank you.
Please change actual in the axis labels for “observed”
R: We have changed the axis label from “actual” to “observed”.
Figure 7 is missing the statistical parameters , units, please improve the figure. Review caption. For example, relationship between predicted and observed xxxx lean n concentration (LNC) and plant height (PH). Authors should check the quality of all figures and captions.
Missing model fitting?
R: Thank you for pointing this. However, based on an observation made by another reviewer, we decided to remove Figure 7 and substitute it for another. Regardless, we agree with the observations made and also checked for the quality of the remaining figures and captions.
Line 229: “estimative” ? -> replace by prediction.
R: We replaced “estimative” with “prediction”.
What are the units of RMSE in figure 4?
R: Thank you for noticing it. They are in g.kg-1.
RR: Please fix the font on the units, “r” -> “pearson’s correlation” . For the caption RMSE has to be explained root mean square…. (RMSE)… ETC.
RRR: We have fixed the font on the units, but the word office is compressing the image quality by default. Because of that, we are submitting a higher resolution of the same image for the editorial office alongside this manuscript. Concerning the caption, we have performed the suggested corrections. Thank you.
Line 194 what is topdressing condition?
R: Thank you for pointing this out. The topdressing condition corresponds with the highest fertilization rate, 180 kg ha-1. We correct it accordantly.
RR: both treatments were top dressing….
RRR: We are deeply sorry for this mistake. We corrected this sentence and also other minor segments were we wrongfully wrote it. In this case, in particular, the new phrase is in lines 209 and 210. We now modified it to be: “The magnitude of the correlations was different for each N fertilization rate (high and low).”
Line 210 – please clarify how they were different the correlations at different top dressing treatments?
“each N content topdressing condition.” => it is not a condition it is a treatment the amount of N , and leaf N is a N status. Please be consistent with the terms.
R: Please see our previous answer. We were referring to different fertilization rates. Thank you again for noticing this. Another large section that this mistake occurred and was also changed was the following:
Lines 266-267: From: “This is an important observation since demonstrates that the RF approach was able to separate well topdressing conditions.”
To: “This is an important observation since demonstrates that the RF approach was able to separate distinctly the low and high rate levels.”
Discussion
Line 283-284 “and …”and” please correct.
R: We corrected it. The RBF and polynomial are kernels from the SVM model. Because of that, it was written in this manner. Regardless we adjusted it to avoid any confusion.
Line 284-285 …the two clusters are the treatment effect. Not clear why the authors are pointing this out in this section of the discussion. And? What are the authors trying to say with this?
“As we perform inter-calibration issues between the different sensors, these results can be attributed to climatic differences” ????? not sure what this means and how this is related with the discussion of a particular result.
R: Indeed, upon evaluating this sentence, we agreed that it is not appropriate nor clearly explained. Because of that, we choose to remove the sentence between lines 286-290. “In some scatterplots in Figure 3, it is possible to notice two clusters formed for some variables. Corn is a C4 plant that increases its plant mass due to the rain that occurred during its cultivation. As we perform inter-calibration issues between the different sensors, these results can be attributed to climatic differences (soil-water-plant-atmosphere interaction) between one crop season and the other (see Figure 2). Besides, our…”.
Line 297 “predict phenotype”. Please demonstrate how are you predicting phenotype…
R: We modify this sentence to “predict height”.
Line 285-286 – is totally out of context…please revise
R: As we stated in our previous response, we choose to remove this phrase to avoid confusion. Thank you for pointing this out.
Line 294 what is plant status? Plant height was used.
R: Thank you for this observation. We correct it accordantly.
RR: what line? Please bring the replies here.
RRR: Unfortunately, we could not find the original lines were this statement was written due to divergence between the manuscript’s versions. However, upon comparing our original manuscript with our current one, there are two phrases that state “status”, and they were already changed in the previous versions. We’ll be listing them here shortly:
Original: “These VIs are among the most commonly implemented indices to evaluate plant health and status conditions.”
Current Lines 152-153: “The mentioned VIs are among the most commonly used indices to predict plant health and conditions.”
Original: “Here, our particular objective was to investigate the contribution of multispectral data into machine learning methods to nutrient content (N) and growth-status (PH).”
Current Lines 360-362: “Here, our particular objective was to investigate the contribution of multispectral data into machine learning methods to nutrient content (N) and height (PH).”
We also looked into every “status” word in the current version of our manuscript to ensure that this mistake did not occur a second time. Thank you.
Line 297 – 299 the paper is focused on 2 variables. The authors did not make the connection with management.
R: Thank you for this observation. We correct it accordantly.
RR: where? Please bring the lines, or section where you address this issue. Thanks
RRR: We remove this sentence from the previous version not only in the 297-299 lines but also in every other section. We’ll be listing here them shortly:
Original (Abstract): “Our method may be adopted in precision agriculture practices and assist proper management and be applied in decision-making models.”
Current Lines 15-16: “Our method may be adopted in precision agriculture practices and assist in decision-making models.”
Original (Discussion): “In this regard, precision agriculture practices could benefit from our framework, supporting agricultural system management.”
Current Lines 366-367: “In this regard, agriculture remote sensing could benefit precision agriculture approaches.”
Original (Conclusion): “Our method may be used in novel research that intends to evaluate different types of crops, or applied in precision agriculture practices and assist proper management and be used in decision-making models.”
Current Lines 379-381: “Our method may be used in novel research that intends to evaluate different types of crops, or applied in precision agriculture practices and assist in decision-making models.”
Are the errors found in the predictions high, low, or medium and how do they affect the use of this tool?
R: Thank you for this question. The errors observed for the RF model can be considered relatively lower or similar to similar research. In this regard, we have added information into our discussion section. Thank you very much.
RR: please bring the lines and part of the MS changed to address this issue. Thanks
RRR: Thank you for bringing this up. To clarify, we choose not to directly compare the prediction values between each manuscript since the only “in-common” procedure with ours was the adoption of the RF model in their method. As they evaluated different plants, sensors, variables, areas, etc. We choose to discuss these implications in the following manner:
Lines 307-309: In similar research [49], an experiment conducted with corn and multispectral imagery from an orbital scale demonstrated that cumulative VIs showed strong performance for the above-ground estimation of biomass.
- Kross, A.; McNairn, H.; Lapen, D.; Sunohara, M.; Champagne, C. Assessment of RapidEye vegetation indices for estimation of leaf area index and biomass in corn and soybean crops. Int. J. Appl. Earth Obs. Geoinf. 2015, 34, 235–248, doi:10.1016/j.jag.2014.08.002.
Lines 309-310: As mentioned, another research [34], aiming to estimate maize biomass, stated that the RF learner returned the highest accuracies among the evaluated algorithms.
Han, L.; Yang, G.; Dai, H.; Xu, B.; Yang, H.; Feng, H.; Li, Z.; Yang, X. Modeling maize above-ground biomass based on machine learning approaches using UAV remote-sensing data. Plant Methods 2019, 15, 1–19, doi:10.1186/s13007-019-0394-z.
Lines 311-314: For N content, although not conducted in corn crops, multiple types of research [26,37,50–53] also concluded that the RF learner, or some other type of regressors based on decision trees, was more appropriate to model LNC. In our approach, the errors encountered with this model are relatively lower or similar when in comparison to the aforementioned studies.
- He, L.; Song, X.; Feng, W.; Guo, B. Bin; Zhang, Y.S.; Wang, Y.H.; Wang, C.Y.; Guo, T.C. Improved remote sensing of leaf nitrogen concentration in winter wheat using multi-angular hyperspectral data. Remote Sens. Environ. 2016, 174, 122–133, doi:10.1016/j.rse.2015.12.007.
- Chlingaryan, A.; Sukkarieh, S.; Whelan, B. Machine learning approaches for crop yield prediction and nitrogen status estimation in precision agriculture: A review. Comput. Electron. Agric. 2018, 151, 61–69, doi:10.1016/j.compag.2018.05.012.
- Varela, S.; Dhodda, P.R.; Hsu, W.H.; Prasad, P.V.V.; Assefa, Y.; Peralta, N.R.; Griffin, T.; Sharda, A.; Ferguson, A.; Ciampitti, I.A. Early-season stand count determination in Corn via integration of imagery from unmanned aerial systems (UAS) and supervised learning techniques. Remote Sens. 2018, 10, doi:10.3390/rs10020343.
- Miphokasap, P.; Wannasiri, W. Estimations of Nitrogen Concentration in sugarcane using hyperspectral imagery. Sustain. 2018, 10, 1–16, doi:10.3390/su10041266.
- Abdel-Rahman, E.M.; Ahmed, F.B.; Ismail, R. Random forest regression and spectral band selection for estimating sugarcane leaf nitrogen concentration using EO-1 Hyperion hyperspectral data. Int. J. Remote Sens. 2013, 34, 712–728, doi:10.1080/01431161.2012.713142.
- Liang, L.; Di, L.; Huang, T.; Wang, J.; Lin, L.; Wang, L.; Yang, M. Estimation of leaf nitrogen content in wheat using new hyperspectral indices and a random forest regression algorithm. Remote Sens. 2018, 10, doi:10.3390/rs10121940.
Lines: 344 Indeed, the combination of multiple variables and configurations implemented in our paper 345 demonstrates a feasible approach to be adopted in this type of analysis. More than often, previous 346 research focused on the evaluation of a single variable our cultivar, in a single period.
RR: These text doesn’t make sense, plus needs English revision. “Indeed” doesn’t fit there…. “More than often”….several places need to be reviewed. “This type of analysis” ….vague argument.
RRR: Thank you for pointing this out. We modified the introduction of this paragraph to the following statement: “The high accuracy obtained by the RF model demonstrates that ML and VIs from UAV-based imagery are a feasible combination for predicting LNC and PH.” We hope that this new version is more appropriate.
Lines”: 352 these conditions are essential to corroborate the robustness of the proposed approach with the 353 machine learning approach. Approach is repeated twice.
R: We changed “with the machine learning approach.” to “with machine learning.”
Conclusions
We also performed an extensive and careful revision of the English language in our text, as well as asking for a native-speaker to evaluate it. We corrected grammar, structure sentences, and other errors that were decreasing the presentation of our manuscript.
RR: please review your new text additions, figures, captions.
RRR: We verify the text, figures, and captions following each statement presented in this letter. Thank you very much for this. We hope that the alterations are suitable for you.
Kind regards,
Authors.
Reviewer 4 Report
Review manuscript remotesensing-864160, "Leaf Nitrogen Content and Plant Height Prediction for Corn-Plants using UAV-Based Multispectral Imagery and Machine Learning Techniques”.
My major issue still concerns the comparison between observed agronomic variables (LNC and PH) and the predicted ones. It is wrong to perform a linear regression between these as shown in figure 6. The only correct way is to compare observed and predicted values by analyzing their difference, so by comparing them with the 1:1 line. In this respect, one may use the RMSE or MAE (both based on the direct comparison between observed and predicted values) for evaluating the results. Calculation of r or R2 after another linear regression is not allowed.
Although the English grammar has been improved somewhat, for me this would still not be acceptable as editor of an international journal. However, I leave this decision up to the editor.
Some examples (but there are more errors in the manuscript):
Line 47: ‘images analysis’ should be ‘image analysis’
Line 48: ‘growth-status estimative’ should be ‘growth-status estimate’
Line 74: full stop missing at end of the sentence
Line 123/124: ‘respectably’ should be ‘respectively’
Line 164: ‘coefficient of regression’ should be ‘coefficient of determination’
Line 366: ‘high’ should be replaced by ‘height’
Line 246 and 377: ‘k-1’ should be ‘kg-1’
A ’;’ often is used incorrectly (and should be replaced by a ‘,’). The semicolon usually combines two parts of a sentence when each of the two parts could form grammatical sentences on their own.
Some final, minor comments:
Figure 2: indicate when 50 DAE occurred.
Figure 6: y-axis of lower left graph seems incorrect, looking at the 1:1 line (axis should go up to 2.8, not 3).
Difference between figure 6 and 7 is not clear, except for the different lay-out. Please explain also in the caption what is shown in the figures.
Author Response
Reviewer 4:
Review manuscript remotesensing-864160, "Leaf Nitrogen Content and Plant Height Prediction for Corn-Plants using UAV-Based Multispectral Imagery and Machine Learning Techniques”.
My major issue still concerns the comparison between observed agronomic variables (LNC and PH) and the predicted ones. It is wrong to perform a linear regression between these as shown in figure 6. The only correct way is to compare observed and predicted values by analyzing their difference, so by comparing them with the 1:1 line. In this respect, one may use the RMSE or MAE (both based on the direct comparison between observed and predicted values) for evaluating the results. Calculation of r or R2 after another linear regression is not allowed.
R: Dear reviewer,
We understand your general concern over the R² graphic matter. In this sense, we added a new Figure which indicates the dispersion of the residual values of the best predictions for both LNC and PH. We believe that this information, alongside the RSME (Figure 4), MAE (Figure 5), r (Figures 4 and 5), and R² (Figure 6) will help the reader to better ascertain the relationship between our variables and judge the proper quality of these results. We hope that these alterations are suitable for you. Thank, again, for your cooperation in this matter.
Although the English grammar has been improved somewhat, for me this would still not be acceptable as editor of an international journal. However, I leave this decision up to the editor.
R: We are sorry that the English corrections did not meet your requirements. Regardless, we performed a new evaluation and checked for any errors aside from the ones punctuated here. We thank you for this meticulous evaluation.
Some examples (but there are more errors in the manuscript):
Line 47: ‘images analysis’ should be ‘image analysis’
Line 48: ‘growth-status estimative’ should be ‘growth-status estimate’
Line 74: full stop missing at end of the sentence
Line 123/124: ‘respectably’ should be ‘respectively’
Line 164: ‘coefficient of regression’ should be ‘coefficient of determination’
Line 366: ‘high’ should be replaced by ‘height’
Line 246 and 377: ‘k-1’ should be ‘kg-1’
A ’;’ often is used incorrectly (and should be replaced by a ‘,’). The semicolon usually combines two parts of a sentence when each of the two parts could form grammatical sentences on their own.
R: We looked into every line and performed the necessary corrections. Thank you.
Some final, minor comments:
Figure 2: indicate when 50 DAE occurred.
Figure 6: y-axis of lower left graph seems incorrect, looking at the 1:1 line (axis should go up to 2.8, not 3).
Difference between figure 6 and 7 is not clear, except for the different lay-out. Please explain also in the caption what is shown in the figures.
R: Figure 2 now has the information regarding when 50 DAE occurred. Figure 6, PH (KNN(1)) y-axis, was corrected. Thank you for noticing this. We choose to remove Figure 7. Thank you.
Kind regards,
Authors.